# Cross-frequency synchronization connects networks of fast and slow oscillations during visual working memory maintenance

Felix Siebenhühner[1], Sheng H Wang[1], J Matias Palva[1], Satu Palva[1,2]*

[1]Neuroscience Center, University of Helsinki, Helsinki, Finland; [2]BioMag laboratory, HUS Medical Imaging Center, Helsinki, Finland

**Abstract** Neuronal activity in sensory and fronto-parietal (FP) areas underlies the representation and attentional control, respectively, of sensory information maintained in visual working memory (VWM). Within these regions, beta/gamma phase-synchronization supports the integration of sensory functions, while synchronization in theta/alpha bands supports the regulation of attentional functions. A key challenge is to understand which mechanisms integrate neuronal processing across these distinct frequencies and thereby the sensory and attentional functions. We investigated whether such integration could be achieved by cross-frequency phase synchrony (CFS). Using concurrent magneto- and electroencephalography, we found that CFS was load-dependently enhanced between theta and alpha–gamma and between alpha and beta-gamma oscillations during VWM maintenance among visual, FP, and dorsal attention (DA) systems. CFS also connected the hubs of within-frequency-synchronized networks and its strength predicted individual VWM capacity. We propose that CFS integrates processing among synchronized neuronal networks from theta to gamma frequencies to link sensory and attentional functions.

*For correspondence: satu.
palva@helsinki.fi

**Competing interests:** The authors declare that no competing interests exist.

## Introduction

Working memory (WM) has a limited capacity of 3–4 objects (*Luck and Vogel, 1997*) and is comprised of sensory storage and central executive control for manipulating the stored information and supporting the preparation of a contingent response (*Baddeley, 1996*; *Miller and Cohen, 2001*; *Sreenivasan et al., 2014*). During visual WM (VWM) maintenance, these functions are achieved in visual and fronto-parietal (FP) brain areas, which exhibit enhanced neuronal activity levels during memory maintenance in both monkey single- and multi-unit recordings (*Buschman et al., 2011*; *Goldman-Rakic, 1995*; *Siegel et al., 2009*) and functional MRI (fMRI) of humans (*Munk et al., 2002*; *Pessoa et al., 2002*; *Todd and Marois, 2004*). While the visual cortex is responsible for the processing of visual information and its maintenance in VWM (*Riggall and Postle, 2012*; *Emrich et al., 2013*; *Kravitz et al., 2013*), lateral prefrontal cortex (LPFC) is thought to regulate VWM maintenance, represent goals and rules, and govern response selection (*Miller and Cohen, 2001*; *Rowe et al., 2000*; *Sreenivasan et al., 2014*; *Markowitz et al., 2015*). Recent studies show that even in primate prefrontal cortex, these VWM functions are achieved in multiple dissociable networks (*Markowitz et al., 2015*; *Lundqvist et al., 2016*).

Phase synchronization (PS) is a mechanism for integrating anatomically distributed processing and regulating the communication between distant assemblies (*Singer, 1999*; *Fries, 2005*, *2015*). Macaque cortical recordings have shown that in beta- (β, 15–30 Hz) and gamma- (γ, 30–90 Hz) frequency bands, inter-areal PS is indeed enhanced by the maintenance of visual

information in VWM and it has been suggested to support the maintenance of VWM content (*Siegel et al., 2009*; *Salazar et al., 2012*). Using MEG and EEG, source-localization, and network analyses (*Bullmore and Sporns, 2009*), we observed that VWM involves large-scale β- and γ-band PS in the visual and parietal regions as in invasive LFP recordings from monkeys, but also high-alpha (α)-band (11–14 Hz) synchronization in fronto-parietal areas (*Palva et al., 2010*). This anatomical pattern suggests a putative functional division where β- and γ-band synchronization underlies the maintenance and integration of visual information while the high-α-band synchronization supports the coordination of processing among brain regions serving attentional and central executive control. Likewise in MEG, also human intracranial EEG (iEEG) data have revealed long-range phase-synchronization in theta (θ, 4–8 Hz), β, and γ bands in hippocampal memory circuits (*Fell et al., 2001*; *Axmacher et al., 2008*; *Burke et al., 2013*).

Similar conclusions have also been made in studies addressing the relationship of VWM performance and oscillation amplitude or power modulations that reflect local cortical synchronization. The amplitudes of both β- and γ-band oscillations are positively correlated with the task performance and individual capacity limits (*Sauseng et al., 2009*; *Palva et al., 2011*; *Roux et al., 2012*; *Honkanen et al., 2015*) and thought to support the neuronal representations of VWM contents (*Tallon-Baudry et al., 1998*; *Honkanen et al., 2015*). In contrast, the plateau of load-dependent strengthening in α-frequency band amplitudes during VWM is negatively correlated with the individual VWM capacity (*Palva et al., 2011*) and α oscillations are stronger for distractors than memorized items (*Bonnefond and Jensen, 2012*; *Park et al., 2014*). Together with a large body of other evidence (*Palva and Palva, 2007*; *Buffalo et al., 2011*; *Saalmann et al., 2012*; *Haegens et al., 2015*), these findings suggest that θ and/or α oscillations underlie the attentional functions of VWM by facilitating the processing of relevant and/or by suppressing irrelevant information (*Jensen et al., 2014*; *Palva and Palva, 2011*).

Hence, a crucial challenge now is to reveal what kinds of cross-frequency coupling (CFC) mechanisms integrate the neuronal processing that during VWM maintenance is distributed to multiple frequency bands in anatomically distinct networks. These mechanisms would conceivably also support the emergence of subjectively coherent VWM from sensory and central executive functions. Phase amplitude coupling (PAC) is one form of CFC that has been suggested to underlie cross-frequency integration (*Palva et al., 2005*; *Palva and Palva, 2007*; *Sirota et al., 2008*; *Canolty and Knight, 2010*; *Fell and Axmacher, 2011*; *Giraud and Poeppel, 2012*; *Lisman and Jensen, 2013*). Prior studies have shown that during VWM maintenance, γ oscillation amplitudes are coupled to θ phase via PAC in cortical (*Sauseng et al., 2009*; *Siegel et al., 2009*) and hippocampal recordings (*Axmacher et al., 2010*; *Park et al., 2016*). However, PAC is independent of the phase of the fast oscillations and hence cannot promote a consistent temporal and spike-time relationship between the slow and fast oscillations (*Palva et al., 2005*; *Fell and Axmacher, 2011*; *Palva and Palva, 2012*), which are essential in the regulation of neuronal communication (*Singer, 1999*; *Fries, 2015*).

In contrast to PAC, *n:m*-cross-frequency phase synchrony (CFS) (*Tass et al., 1998*; *Palva et al., 2005*; *Palva and Palva, 2007*) is a form of CFC that indicates direct phase coupling of the slow and fast oscillations. CFS could thus mechanistically underlie the cross-frequency integration of synchronized networks at the temporal resolution of the faster oscillation (Fell and Axmacher, 2011; *Palva and Palva, 2012*) through coordination of consistent spike timing relationships between the oscillations. In CFS, the ratio of high ($f_{high}$) and low ($f_{low}$) frequencies is defined by the integers $n$ and $m$ so that $n \cdot f_{high} = m \cdot f_{low}$. In humans, CFS has been observed in resting-state (*Nikulin and Brismar, 2006*) and VWM tasks among EEG sensors (*Schack et al., 2005*; *Hamidi et al., 2009*; *Sauseng et al., 2009*) and in a mental calculation WM task among MEG sensors (*Palva et al., 2005*). However, being limited to the sensor level, these studies leave the cortical sources of CFS as well as its functional significance unclear. A recent study using intracranial EEG (iEEG) also reported behaviorally significant θ:β and θ:γ CFS during VWM in human hippocampus using a serial Stenberg task (*Chaieb et al., 2015*). Additionally, CFS has also been observed in the neocortical (*Roopun et al., 2008*) and hippocampal (*Belluscio et al., 2012*) microcircuits in rats.

We postulate that CFS supports the integration of neuronal processing between synchronized networks in distinct frequency bands during VWM maintenance and thereby underlies the integration of sensory and attentional functions of VWM (*Figure 1*). We use here concurrent magneto- and electroencephalography (M/EEG) with individual-anatomy-based source reconstruction and data-driven analyses to investigate the functional significance of CFS in WVM. The key predictions of our

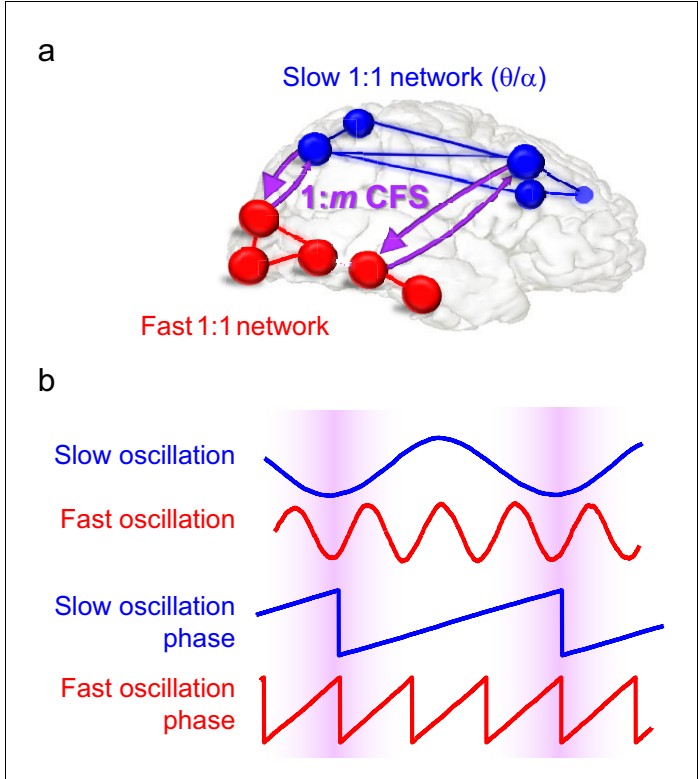

**Figure 1.** Schematic illustration of the hypothesis. (**a**) Fast gamma (γ) network synchronization (red) integrates processing within the visual system while slow theta and/or alpha (θ/α) synchronization (blue) supports attentional processing in the fronto-parietal network. Cross-frequency phase synchronization (CFS) (purple) connects the fast and slow networks and regulates their communication to integrate sensory and attentional functions. (**b**) 1:3-phase synchronized slow (blue) and fast (red) oscillations and their phase time series. Highlights indicate how cross-frequency phase synchronization endows the fast and slow oscillation time windows where their systematic phase relationship allows spike coincidence detection and other spike-time dependent interactions to take place.

postulate are that CFS should be observable and load-dependent in a parametric VWM task (*Palva et al., 2010*), that it should connect the task-relevant within-frequency-synchronized networks, and finally that it should be predictive of individual behavioral VWM performance. We found solid support for each of these predictions, which shows that CFS may indeed be a mechanism for coordinating the neuronal processing and regulating communication among oscillatory networks at different frequencies.

## Results

### Inter-areal CFS is a robust phenomenon during VWM retention period

We assessed here the functional role of CFS in VWM by analyzing M/EEG data acquired during a delayed match-to-sample VWM task (*Luck and Vogel, 1997*; *Palva et al., 2010*, *2011*), which had earlier revealed concurrent large-scale 1:1 synchronization in α-, β-, and γ-frequency bands (*Palva et al., 2010*). In each trial of the task, the subjects were presented a to-be-memorized 'Sample' stimulus containing one- to six-colored squares and, after a memory retention period of 1 s, a 'Test' stimulus, wherein 50% of trials one square was of a different color than in the Sample (*Figure 2a*). The subjects gave a forced-choice response about whether the Test was same or different than the Sample. The hit rate (HR) decreased as a function of VWM load but remained above chance level for the highest load of six objects (*Figure 2—figure supplement 1*).

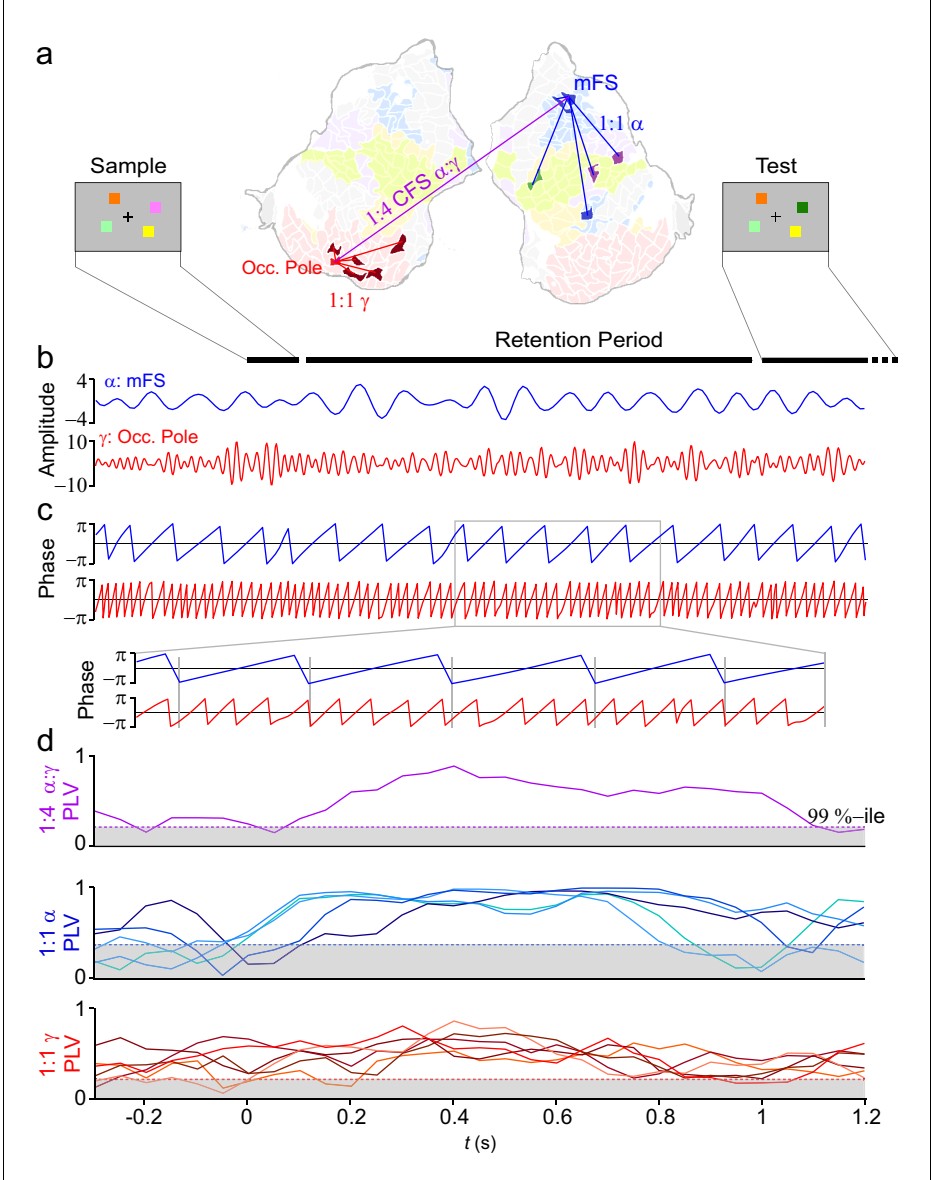

**Figure 2.** VWM modulates inter-areal cross-frequency phase synchrony (CFS) in human cortex. Example of prominent single-trial CFS during the VWM retention period. (a) Illustration of the delayed-match-to-sample experimental paradigm where the Sample stimulus containing 1–6 colored squares is presented for 0.1 s and then after a 1 s retention period a Test stimulus appears and the subject responds whether the Test contains an object with a color different from than in Sample. The flattened cortical surface shows an example of 1:4 CFS (purple line) transiently phase coupling α (13 Hz) oscillations in the right medial frontal sulcus (MFS) with γ (54 Hz) oscillations in left occipital pole (Occ.Pole) during VWM retention. These α and γ oscillations were concurrently also 1:1 synchronized in networks of other cortical areas (blue and red lines, respectively). The cortical surface is colored by brain systems as indicated in *Figure 4.* (b) The signals from right MFS and left occ. pole were filtered with Morlet wavelets at 13 Hz and 54 Hz respectively. (c) The phase of the signals in (b). The zoom-in insert illustrates visually salient 1:4 cross-frequency phase synchrony during the retention period. (d) Top panel: 1:4 CFS between these α and γ oscillations was quantified with the phase-locking value (PLV) in sliding 300 ms time windows. The grey area denotes the 99%-ile confidence intervals of null hypothesis PLV values obtained with time-shifted surrogate data. CFS was strengthened during VWM retention significantly above the baseline and null hypothesis levels. Middle panel: 1:1 α phase synchrony between MFS and the five connected fronto-parietal parcels shown in (a) Bottom panel: 1:1 γ phase synchronization of Occ. Pole with the other visual cortex areas shown in (a).

The following source data and figure supplements are available for figure 2:

*Figure 2 continued on next page*

*Figure 2 continued*

**Source data 1.** List of parcels.
**Figure supplement 1.** Behavioral performance.
**Figure supplement 2.** Parcellations and subsystems.

We estimated the phase time series of single-trial ongoing neuronal activity for all cortical grey matter sources by filtering artefact-cleaned M/EEG recordings with Morlet wavelets ranging from 3 to 90 Hz in frequency and using cortically-constrained minimum-norm estimates (*Lin et al., 2006*) for the inverse transform. These source estimates were collapsed with individually optimized collapse operators (*Korhonen et al., 2014*) to neuroanatomically labeled cortical parcellations of 400 parcels (see *Figure 2—figure supplement 2*).

To first re-illustrate with real data the concept (see *Figure 1*) of CFS connecting cortical parcels engaged in 1:1 synchronization in α and γ bands, we selected two cortical parcels in a single subject: middle frontal sulcus (mFS) in lateral prefrontal cortex (LPFC) for α oscillations and occipital pole (early visual cortex) for γ oscillations (*Figure 2b*), and estimated their phase time series (*Figure 2c*). We then quantified n:m CFS (*Palva et al., 2005*; *Tass et al., 1998*) and 1:1 synchronization with the phase-locking value (PLV) in sliding time windows (*Figure 2d*) and found in single trials that CFS between these regions as well as 1:1 α and γ band synchronization in separate networks were strengthened during VWM retention.

To characterize this phenomenon at the group level (*N* = 12), we quantified event-related CFS obtained across trials for each pair of frequency bands and each pair of the 400 cortical parcels. Mapping the correlation networks at the whole-brain level is crucial for understanding neuronal communication in cognitive architectures (*Petersen and Sporns, 2015*). As the n:m ratios, we used *n* = 1 and *m* = 2−9 and assessed CFS in two pre-stimulus and two VWM-retention period time windows (0.4−0.7 and 0.7−1 s from Sample onset). The retention period windows were free of stimulus-evoked activity (*Palva et al., 2011*), which is a major confounder in analyses of both inter-areal PS and all forms of CFC (*Palva and Palva, 2012*). For robust statistical mapping of CFS at the group level, we collapsed the parcel-parcel adjacency matrices of CFS strength (PLV) to a coarser parcellation, the Destrieux atlas (*Destrieux et al., 2010*), where the cortical surface is divided into 148 parcels (see *Figure 2—figure supplement 2* and *Figure 2—source data 1*). The workflow for obtaining the results shown in *Figures 3−9* is shown in *Figure 3—figure supplement 1*.

We first addressed whether the inter-areal CFS was strengthened or suppressed during VWM retention compared to the pre-Sample baseline (two-sided *t*-test, α = 0.05, corrected). We initially averaged the CFS estimates across the six memory-load conditions ('Mean condition') and represented significant differences in CFS between baseline and retention as graphs where cortical areas comprise the vertices and significant connections are edges (*Bullmore and Sporns, 2009*). These task effects on the CFS were indexed at the graph level by connection density, *K,* that was the proportion of statistically significant inter-areal interactions of all possible pair-wise interactions among the 148 brain areas. Visualizing *K* as a function of $f_{low}$ and the frequency ratio 1:*m* revealed that inter-areal CFS of high-α (11–14 Hz) with β- and γ-band oscillations as well as CFS of high-θ-band (5−7 Hz) with α-, β- and γ-band oscillations were enhanced above baseline levels for most or all frequency ratios (*Figure 3a*, see external source data 1 for statistical information, *Siebenhühner et al., 2016*). In contrast, CFS of low-θ (3−4 Hz) and low-α (7−10 Hz) oscillations with their corresponding higher frequency bands was concurrently suppressed below baseline levels (*Figure 3b*; external source data 1, *Siebenhühner et al., 2016*). These data hence show that CFS among cortical oscillations is both up- and down-regulated dynamically by VWM maintenance. The horizontal concentration of positive and negative findings across the frequency ratios to consistent lower frequencies (see *Figure 3a and b*) suggested that the high-α and θ oscillations were synchronized in a harmonic structure with higher frequency oscillations in α−γ bands. We quantified this by comparing the harmonic consistency of CFS across frequency ratios against surrogate data (see Materials and methods) and found this consistency to be significant for the enhanced CFS in high-θ and high-α bands and for the suppressed CFS in the low-α band (*Figure 3a and b*, *p* < 0.05, FDR corrected).

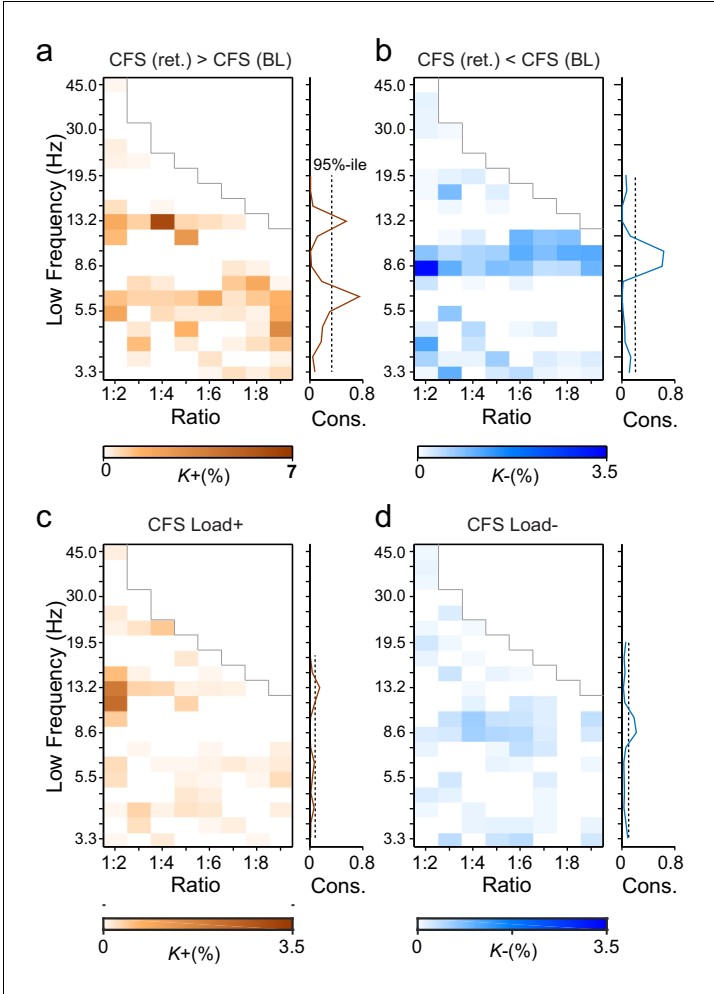

**Figure 3.** During VWM retention, inter-areal CFS was both strenghtened and suppressed in harmonic structures. (a) Fractions ($K+$) of inter-areal CFS connections that were significantly stronger during VWM maintenance than in the pre-stimulus baseline (Mean condition, Wilcoxon signed ranked test, $p < 0.05$, corrected). $K+$ values (color scale) are shown for all studied pairs of $1:m$ ratios from 1:2 to 1:9 (x-axis) and lower frequencies from 3.3 to 45 Hz (y-axis) and represent data from an adjacency matrix for each frequency-pair. The grey line marks the boundary set by the highest investigated frequency (90 Hz). CFS of high-θ and high-α with their upper frequencies was increased for essentially all ratios. Right: The brown line indicates the harmonic consistency of CFS across low frequencies. The dashed line denotes the 95%-ile confidence limit. CFS of high-θ and high-α oscillations with their harmonics at higher frequencies is significantly consistent across ratios. (b) Fractions of inter-areal CFS connections ($K-$) that were suppressed below baseline levels during the retention period. Harmonic consistency (blue line) was estimated as in and shows that low-α consistent CFS was suppressed at all ratios.(c) Fractions of CFS connections that were significantly positively ($K+$) or (d) and negatively ($K-$) correlated with VWM load (Spearman Rank correlation test of CFS across the six VWM memory load conditions, $p < 0.05$, corrected).

The following figure supplements are available for figure 3:

**Figure supplement 1.** Workflow.

**Figure supplement 2.** Leave-one-out statistics and effect size thresholding corroborate the robustness of CFS observations.

**Figure supplement 3.** Apparent signal-to-noise ratio (aSNR) in source space.

**Figure supplement 4.** CFS PLV changes in the Mean condition are not attributable changes in the signal-to-noise ratio (SNR).

*Figure 3 continued on next page*

*Figure 3 continued*

**Figure supplement 5.** Observed PLV changes in the Load condition are not predicted by changes in SNR.

**Figure supplement 6.** Changes in PLV and amplitude are not correlated.

**Figure supplement 7.** Cross-frequency (CF) amplitude-amplitude correlations do not have a low-frequency consistent harmonic structure.

**Figure supplement 8.** Connections with significant CFS are not co-localized with those exhibiting significant CF amplitude correlations.

These data thus suggest that θ and high-α oscillations are systematically phase coupled with β and γ networks for which they thus provide temporal framing across a range of frequency ratios.

To then assess the relationship of CFS with progressive VWM task demands, we asked whether the strength of CFS was systematically correlated with the memory loads from 1 to 6 objects ('Load condition'). We found that CFS between the high-α-band consistent frequency pairs was positively correlated with memory load (Spearman's rank correlation test, α = 0.05, false discovery rate (FDR) corrected) while for θ-consistent CFS, VWM load-dependent modulations were weak. Further, low-α consistent CFS was negatively correlated with VWM load (*Figure 3c–d*; external source data 1, *Siebenhühner et al., 2016*).

To consolidate these observations and ensure that they were not driven by outliers or attributable to effects that would be unreliably sampled at the current cohort size, we performed two lines of control statistics for positive observations in the mean and load conditions. First, using a leave-one-out approach, we created 12 datasets of 11 subjects and performed the group statistics as earlier. We then examined each significant parcel-parcel CFS observation of the prior group statistics (see *Figure 3a–d*) and considered significant only those connections that were significant in *every* leave-one-out cohort. Second, we estimated the effect size that could be estimated at statistical power 1−β of 0.8 with the original cohort size and then from the results of the original group statistics accepted as significant only observations exceeding this limit. Neither the leave-one-out (*Figure 3— figure supplement 2a*) nor the effect size (*Figure 3—figure supplement 2b*) masks or their combined effects (*Figure 3—figure supplement 2c*) led to marked changes in the overall patterns of Mean or Load (*Figure 3—figure supplement 2d–f*) condition findings. The present findings are thus robust despite the smallish sample size and cannot be attributable to any single subject.

## CFS is not attributable to concurrent event-related amplitude changes

The results so far showed that CFS is a robust, dynamic, and task-dependent phenomenon in human cortical activity and is, both theoretically and in the light of the present empirical data, a plausible mechanism for the integration of neuronal processing distributed across frequencies. However, event-related changes in CFS could be attributable to changes in the signal-to-noise ratio (SNR) caused by task-induced changes in oscillation amplitudes. To rule out this possibility, we obtained source-space parcel-specific estimates of apparent SNR (aSNR) (*Figure 3—figure supplement 3*) and used the event-related and memory-load dependent changes in aSNR to predict the changes in CFS estimates that one would observe in the absence to true changes in phase coupling (*Palva et al., 2010*) see Materials and methods). However, we did not find any systematic relationship between SNR and CFS, and observed that as in similar prior analyses for 1:1 PS (*Palva et al., 2010*), the observed CFS changes were much larger than those predicted by changes in SNR in both the Mean (*Figure 3—figure supplement 4*) and Load (*Figure 3—figure supplement 5*) conditions. To further corroborate that the observed CFS effects were not attributable to SNR changes in a case where the SNR would be overestimated, we quantified the correlation between the Mean- and Load-condition effects in PLV and oscillation amplitudes for each significant edge, for each ratio, and for each frequency-pair (see Materials and methods). In line with the prior findings, we found that the task effects on oscillation amplitudes were either insignificantly correlated with the CFS effects or, in the few ratio-frequency pairs where a significant correlation was observed, explained

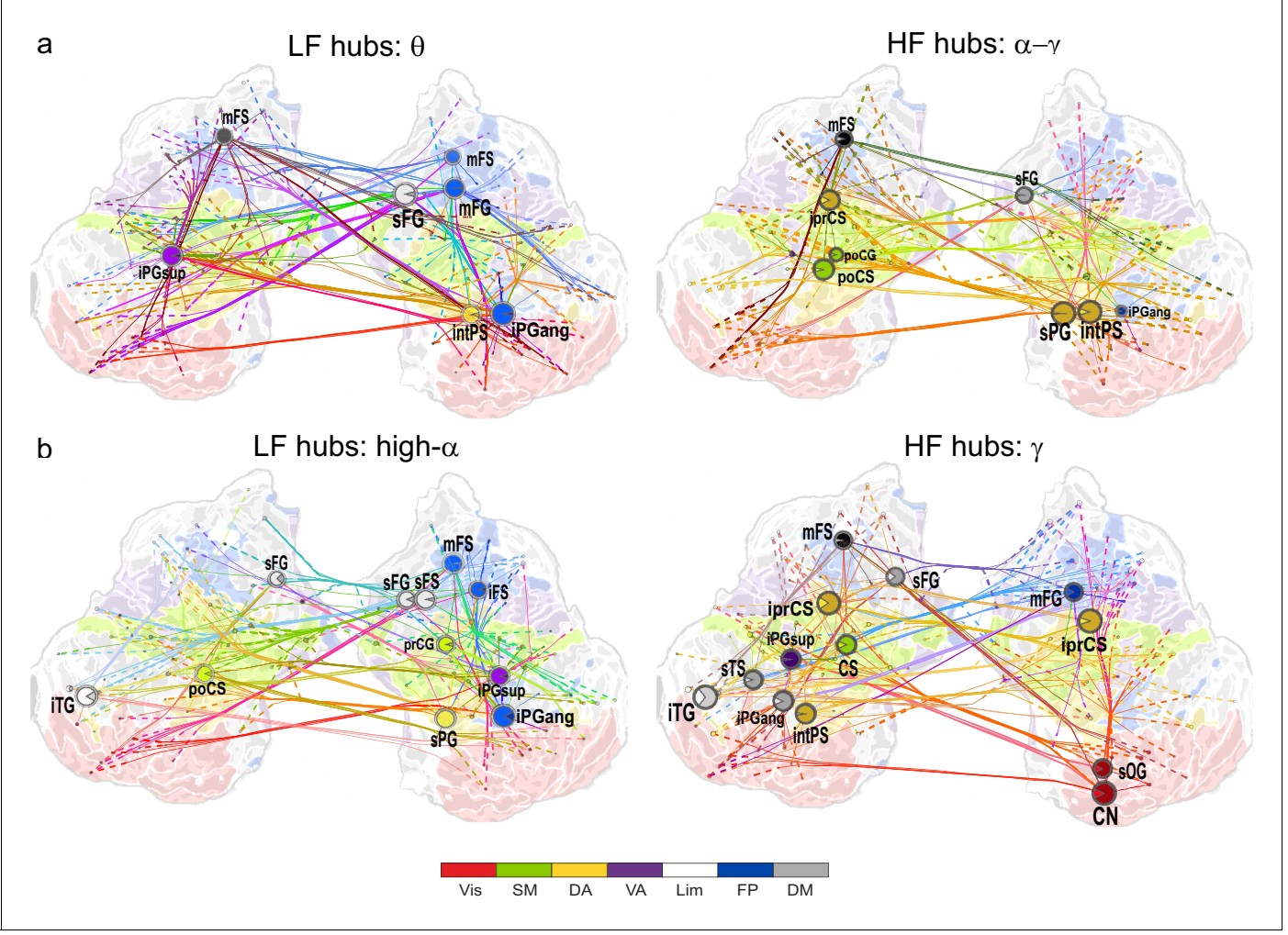

**Figure 4.** θ- and high-α-consistent CFS is observed in fronto-parietal and visual networks. (a) Low-frequency CFS hubs (LF, left) and high-frequency CFS hubs (HF, right) and their most central connections during the VWM retention period for θ-consistent CFS averaged across memory loads (Mean condition) and shared across frequency ratios from 1:2−1:9 (see Materials and methods for details) displayed on a flattened cortical surface. These networks and the most central low- and high-frequency hubs are presented here separately to illustrate the distinct neuroanatomical sources of the most central cortical θ oscillations (left) and the α−γ oscillations (right) connected by this CFS coupling. The complete θ-consistent CFS network is thus the combination of these graphs. Circles represent network vertices (cortical parcels), their size is proportional to their degree centrality, and both vertex and surface colors represent functional brain systems (see colors below). The lighter sector of each vertex represents the fraction of low-frequency connections and the darker section the fraction of high-frequency connections this vertex has in the graph. The width of the edges represents the coupling strength between the two vertices and their color is a mixture of the brain system assignments of the vertices. The dotted lines at the end of the connections point towards the high-frequency (from LF hubs, left panel) or the low-frequency (from HF hubs, right panel) vertices for LF and HF networks, respectively. (b) As in (a), but for the high-α-consistent CFS shared across ratios from 1:3−1:9. Functional subsystem abbreviations: Vis: Visual, SM: Sensorimotor, DA: Dorsal Attention, VA: Ventral Attention, Lim: Limbic, FP: Frontoparietal, DM: Default Mode.

The following figure supplements are available for figure 4:

**Figure supplement 1.** 1:2 CFS between high-α and β bands is most pronounced in the sensorimotor network.

**Figure supplement 2.** Network of low-α consistent CFS suppression.

only a small fraction of the variance in CFS PLV (*Figure 3—figure supplement 6*). Overall, the pattern of these correlations was far from the pattern of CFS and hence the Mean- and Load-condition CFS effects are unlikely to be explained by concurrent changes in SNR.

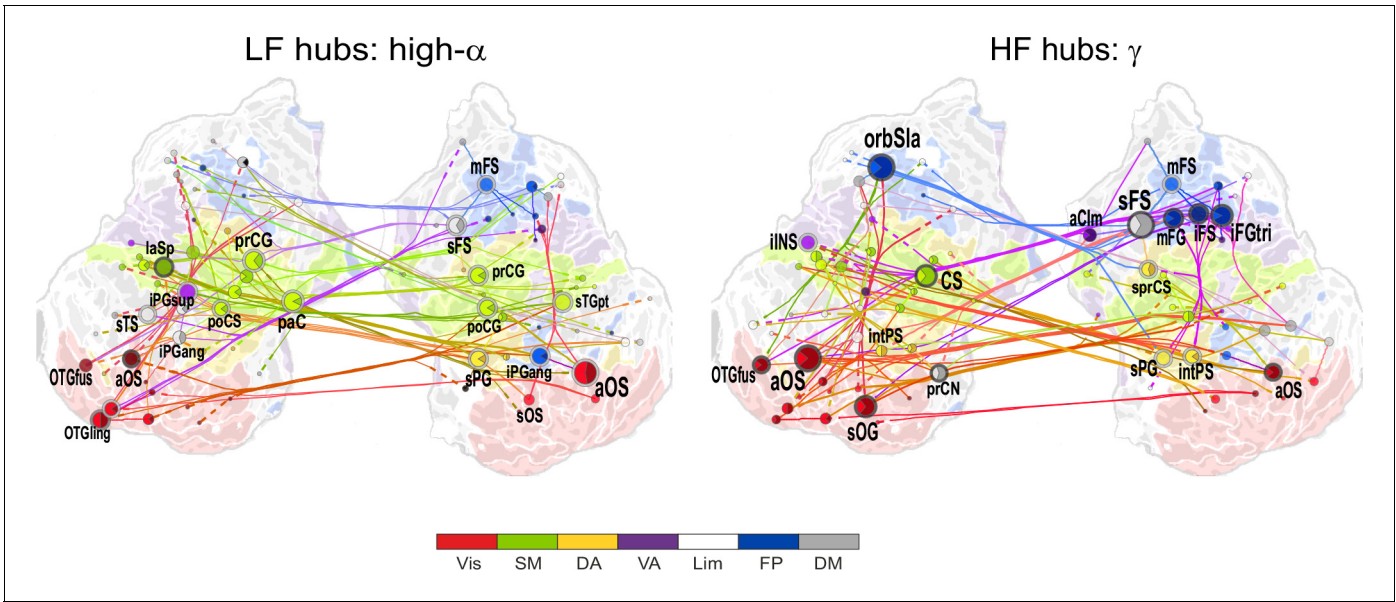

**Figure 5.** 1:3−1:9 CFS between high-α and γ oscillations is VWM load-dependently strengthened among visual, sensorimotor, and frontoparietal networks during the retention period. Low-frequency high-α (LF, left) and corresponding high-frequency γ (HF, right) CFS hubs and their connections for the Load condition high-α-consistent CFS shared across ratios 1:3−1:9 (all illustration details as in *Figure 4a*).

The following figure supplement is available for figure 5:

**Figure supplement 1.** VWM-load-dependent strengthening of 1:2 CFS between high-α and β oscillations is observed in largely in sensorimotor, but also in attentional brain systems.

To further test whether event-related changes in non-sinusoidal/spike-like waveforms (*Palva et al., 2005*) could have led to artificial observations of CFS, we estimated cross-frequency amplitude-amplitude correlations. For CFS attributable to non-sinudoidal waveforms, these amplitude correlations should exhibit a frequency-ratio pattern similar to that of CFS. However, the observed pattern of CF amplitude correlations was saliently distinct from CFS (*Figure 3—figure supplement 7*; external source data 2, *Siebenhühner et al., 2016*) and within each frequency-ratio pair, the vast majority of significant CFS connections did not exhibit significant amplitude-amplitude coupling (*Figure 3—figure supplement 8*). These analyses thus rule out the possibility that changes in SNR or in waveform non-sinusoidalities could have contributed significantly to the observations of CFS in this study.

## CFS during VWM retention links visual and attentional systems

If CFS supports the integration of spectrally distributed processing of attentional and representational functions during VWM, it should be localized to the task-relevant visual and attentional brain systems, and the strength of CFS in these regions should be predictive of behavioral performance. To illustrate the most robust CFS networks in these data, we identified the connectivity backbones of high-α- and θ-consistent CFS networks across the frequency ratios 1:2−1:9 by pooling the adjacency matrices of significant connections, rejecting connections that were significantly attributable to any single ratio, and visualizing the most central connections and network hubs (see Materials and methods).

As CFS networks are asymmetric, the visualization was performed separately for the low- and high-frequency parts of CFS networks. In the visualization, we also accounted for the M/EEG signal mixing effects with a novel clustering approach that bundles the connections more likely to reflect a single true underlying neuronal interaction (see Materials and methods). The brain regions putatively supporting attentional and other functions were identified by co-localizing our parcellation with the

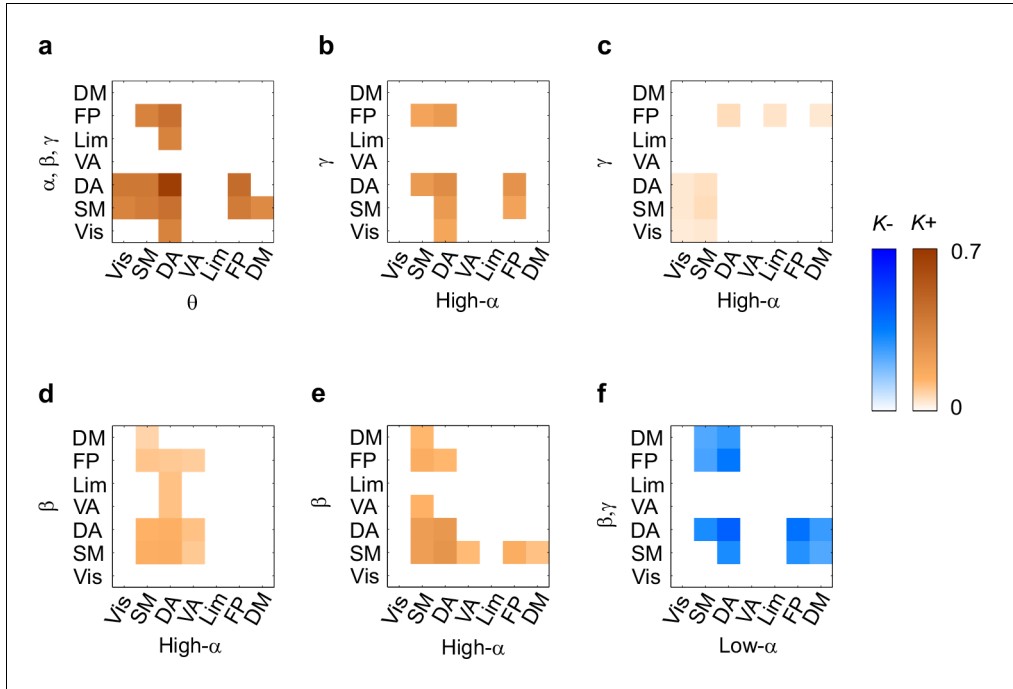

**Figure 6.** CFS connects task-relevant functional brain systems. (a) Connection density *K* for θ consistent CFS (ratios 1:2–1:9) in the Mean condition within and between functional brain systems of the Yeo parcellation. The *x*-axis shows the subsystems for the lower frequency and the *y*-axis for the higher frequency oscillations. *K* is shown for those brain system pairs where *K* exceeded the 95%-ile of the corresponding *K* values obtained with surrogate data. (b,c) Same as (a), but for high-α consistent CFS (ratios 1:3–1:9) in the Mean and Load conditions, respectively. (d,e) Same as (a), but for α:β CFS (ratio 1:2) in the Mean and Load conditions, respectively. (f) *K* of low-α consistent CFS decreases (negative Mean condition) within and between systems.

The following source data is available for figure 6:

**Source data 1.** Statistical table for connection densities.

boundaries of seven predominant brain systems defined by the modularity of intrinsic BOLD signal correlations in fMRI (*Yeo et al., 2011*) (see *Figure 2—figure supplement 2*).

We first visualized the most central hubs and connections of θ-harmonic networks for ratios from 1:2 to 1:9 in the Mean condition separately for the low-frequency (LF, here θ) and high-frequency (HF, here α−γ) parts of the CFS interactions. LF network hubs, *i.e.*, areas wherein the θ oscillations were coupled with α−γ oscillations elsewhere, were found mainly in the fronto-parietal (FP), dorsal (DA), and ventral (VA) attention systems and included the right-hemispheric middle frontal gyrus and sulcus (mFG, mFS), intraparietal sulcus (intPS), and bilateral inferior parietal gyri (iPG) (see large circles in *Figure 4a*, left; external source data 3, *Siebenhühner et al., 2016*). These θ hubs were CFS-coupled with HF oscillations in visual, parietal, and frontal cortices (see dashed ends of connections). HF hubs, *i.e.*, loci where α−γ oscillations were predominantly coupled with θ elsewhere, were observed in the DA system, namely in the left inferior precentral sulcus (iprCS), right superior parietal gyrus (sPG), and intPS as well as in postcentral sulcus and gyrus (poCS/G) of the sensorimotor system (SM) and superior frontal gyrus (sFG) of the default-mode (DM) system (*Figure 4a*, right).

We next visualized the most central hubs and connections of the high-α-CFS network with the same approach. For the CFS between high-α and γ, in networks with ratios 1:3−1:9, we observed LF (high-α) hubs in the right-hemispheric iPG, mFS, and iFS in the FP system (see blue hubs in *Figure 4b*, left panel) and in bilateral sFG, right superior frontal sulcus (sFS), and inferior temporal gyrus (iTG) in the DM system. Among other targets, the FP α-hubs were bilaterally and strongly connected to γ oscillations in the visual cortex (see dashed line ends of red FP-Vis connections in *Figure 4b*, left panel). The HF (here, γ) hubs included mFG and mFS (FP), bilateral iprCS and left

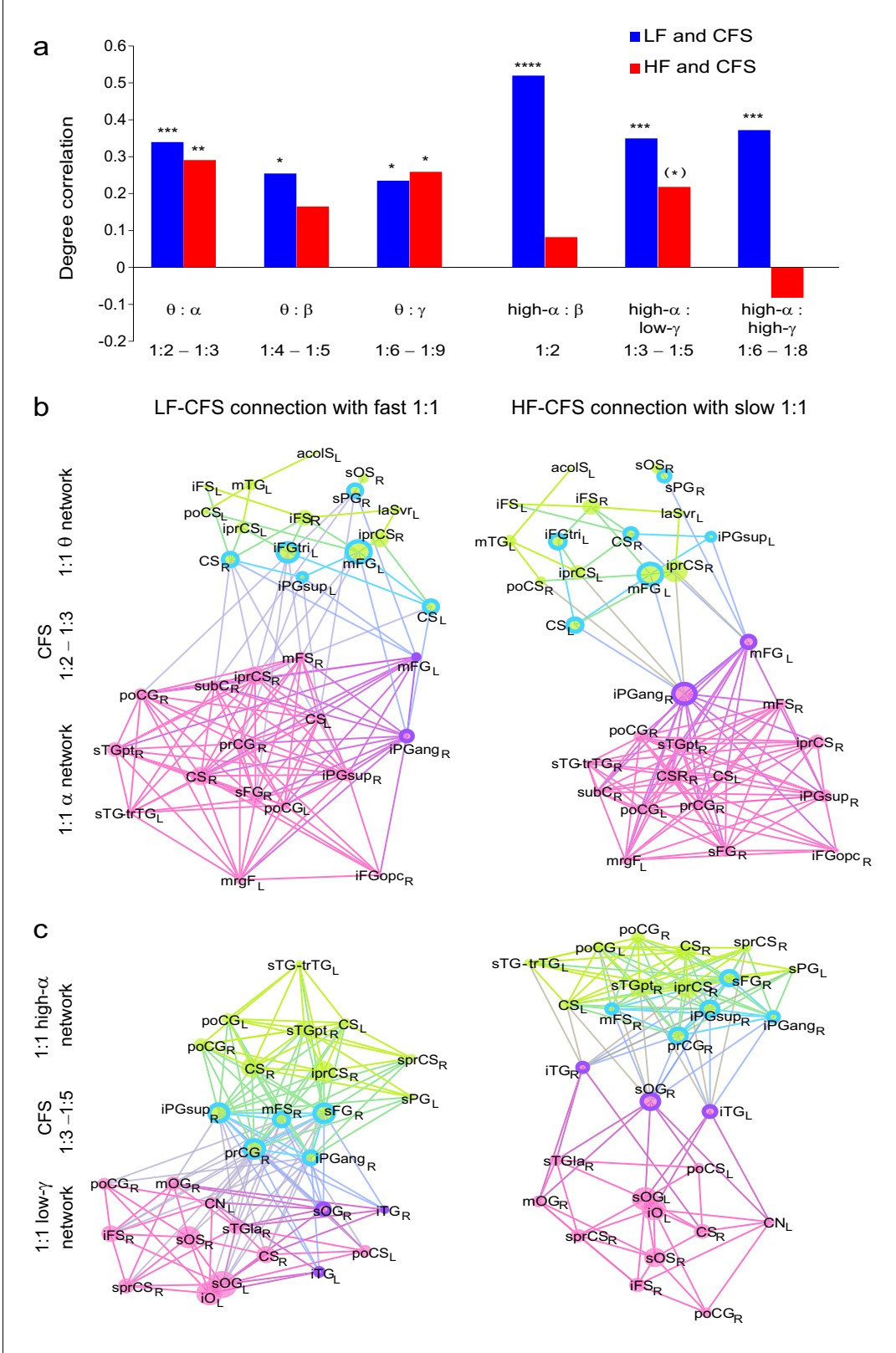

**Figure 7.** CFS connects the hubs of the low- and high-frequency 1:1 phase synchronized networks. (a) Correlation of vertex degree centrality values of the low- (blue) and high- (red) frequency CFS network vertices with the vertices of 1:1 phase synchronized networks in the same frequency bands (Pearson correlation test, \*\*\*$p < 0.001$; \*\*$p < 0.01$; \*$p < 0.05$; (\*) $p < 0.05$ uncorrected. The $p$ values, save for (\*), are corrected for multiple comparisons

*Figure 7 continued on next page*

*Figure 7 continued*

with Benjamini-Hochberg FDR method). (**b**) Simplified illustration of CFS connecting 1:1 networks. The graph shows 15 brain areas with the greatest degree centrality in significantly 1:1 phase synchronized (Mean-condition) θ-band (green) and α-band networks (pink). Blue borders indicate the brain areas that belong to the most central low-frequency CFS-hubs while the violet borders indicate the brain areas that are among the most central high-frequency CFS hubs. Significant CFS coupling between these networks is shown in a separated manner for clarity: left column shows CFS coupling of the low-frequency CFS hubs with all of their targets in the 1:1 higher-frequency network. Conversely, the right column shows the CFS coupling of the high-frequency CFS hubs with all of their targets in the 1:1 lower-frequency network. Individual brain areas may appear both as low- and high-frequency hubs within the same graph. (**c**) CFS coupling between 1:1 high-α-band network (green) and the low-γ-band network (pink). All visualization details as in (**b**).

The following source data and figure supplements are available for figure 7:

**Source data 1.** Statistical table for correlation coefficients r and p-values.

**Figure supplement 1.** CFS connects the hubs of 1:1 synchronized networks in θ and β-γ bands.

**Figure supplement 2.** CFS connects the hubs of 1:1 synchronized networks in α and β−high-γ bands.

**Figure supplement 3.** Network vertex centralities are correlated among oscillations with nearby frequencies.

---

intPS in the DA system and, importantly, the right-hemispheric cuneus (CN) and superior occipital gyrus (SOG) in the visual system. These visual system γ hubs were connected to high-α-oscillations in LPFC and with left-hemispheric posterior parietal cortex (PPC) and visual system (*Figure 4b*, right). CFS between high-α and γ hence connected the FP and DA internally and these attentional systems with the visual system. Moreover, in both the θ- and high-α-consistent CFS networks (see *Figure 4a and b*), the LF hubs were more salient in right-hemispheric FP areas while HF hubs were more prominent in bilateral DA and visual areas. This tentative dichotomy is well in line with the idea that θ/α oscillations serve higher attentional/executive functions (FP) while γ oscillations serve visual representational and lower-level attentional (DA) functions. Both LF and HF hubs were also observed in DM and most commonly in sFG and sFS therein.

In contrast with the 1:3−1:9 α:γ CFS, 1:2 CFS connections between high-α and β oscillations were saliently localized to the SM, DA, and VA systems (*Figure 4—figure supplement 1*; external source data 5, *Siebenhühner et al., 2016*) and were thus visualized separately from α:γ CFS. The differences between α:β and α:γ CFS suggest that they may underlie functionally distinct CFC.

To localize the suppression of CFS and address whether it was a characteristic of the task–relevant or -irrelevant circuitry, we visualized the hubs and connections for the low-α consistent CFS networks that exhibited the greatest suppression during VWM maintenance. We found that both LF and HF hubs were predominantly found in the DM and SM systems that are putatively task-irrelevant in VWM (*Figure 4—figure supplement 2*; external source data 6, *Siebenhühner et al., 2016*). However, a LF low-α hub was also observed in intPS (DA) and HF β / γ hubs in iprCS (DA) and mFG (FP), which were connected to the nodes in the DM system.

We then visualized the most central connections and nodes of the high-α consistent networks that were positively correlated with VWM load ('Load condition'). High-α (LF) hubs were mostly found in the left-hemispheric visual system, right-hemispheric FP, including the mFS, and iPG, and in SM bilaterally (*Figure 5*; external source data 4, *Siebenhühner et al., 2016*). The HF γ hubs were observed in the areas overlapping with those of high-α hubs in the left-hemispheric visual system and in right-hemispheric mFG and sPG. Interestingly, mFS and iFS were key frontal HF nodes and were connected to the visual system similarly to what was observed in the Mean condition α(LF) networks. This suggests that the reciprocality of α and γ oscillations in these areas is enhanced with increasing VWM load. Further, similarly to the Mean condition, also the load-dependent 1:2 CFS connections between high-α and β oscillations were localized to the SM, DA and VA networks (*Figure 5—figure supplement 1*; external source data 5, *Siebenhühner et al., 2016*).

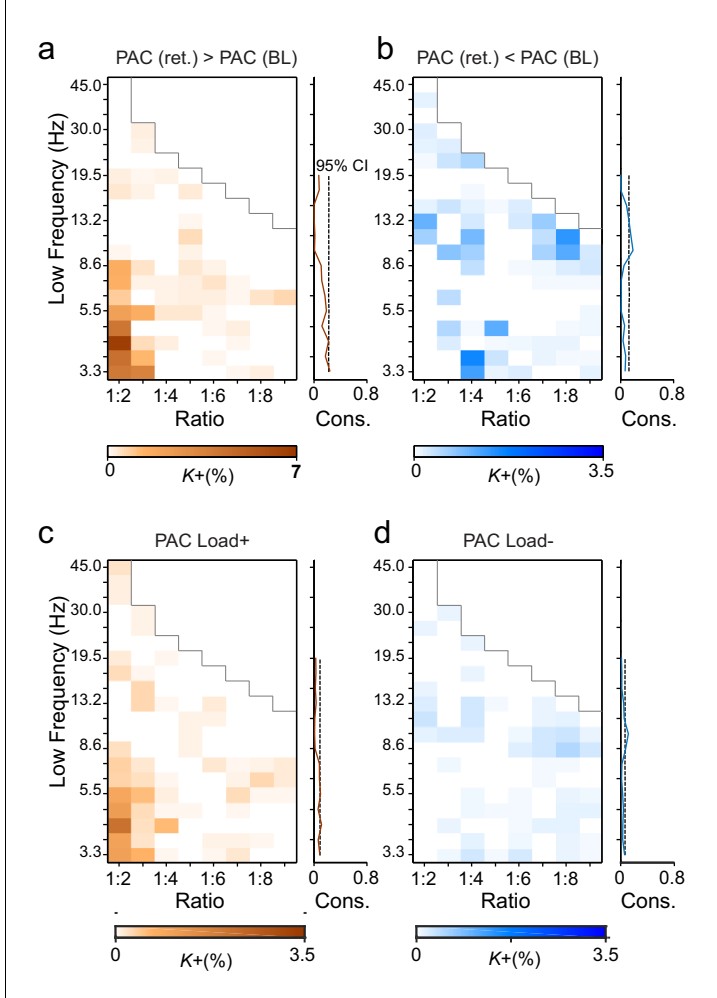

**Figure 8.** Phase-amplitude coupling (PAC) couples the phases of low-frequency θ and α oscillations with the amplitudes of faster α–γ oscillations. (**a**) Fractions of strengthened (*K*+) and (**b**) suppressed (*K*-) PAC in the Mean condition. PAC couples the phase of the low-frequency (y-axis) oscillations with the amplitudes of the faster oscillations at the frequency ratio on the x-axis. The harmonic consistency for PAC is shown on the right as in *Figure 3a and b* for CFS. PAC was suppressed below baseline levels for the high-α consistent ratios where CFS was strengthened and did not show significant harmonic consistency. (**c**) Load-dependent increases and (**d**) decreases in PAC (comparable with the Load condition CFS in *Figure 3c and d*).

The following figure supplement is available for figure 8:

**Figure supplement 1.** Leave-one-out statistics and effect size thresholding corroborate also the robustness of PAC observations.

## Brain-system-level connectivity analysis of the CFS effects

To further elucidate CFS connectivity among brain systems, we complemented the detailed network visualization of the strongest CFS connections (see *Figures 4* and *5*) by assessing the densities of all significant CFS connections, *K*, within and between the seven functional brain systems. These *K* values were compared against non-parametric statistical thresholds obtained by shuffling the adjacency matrices. The system-system connection density matrices largely corroborated the main phenomena found in the network visualizations and showed that VWM retention was associated with significantly denser CFS coupling of θ-oscillations in DA, FP, visual, SM, and DM systems with α−γ oscillations in SM and DA than expected by chance (*Figure 6a*, see *Figure 6—source data 1*). θ-oscillations in DA were also synchronized with α−γ oscillations in visual, FP, and limbic systems.

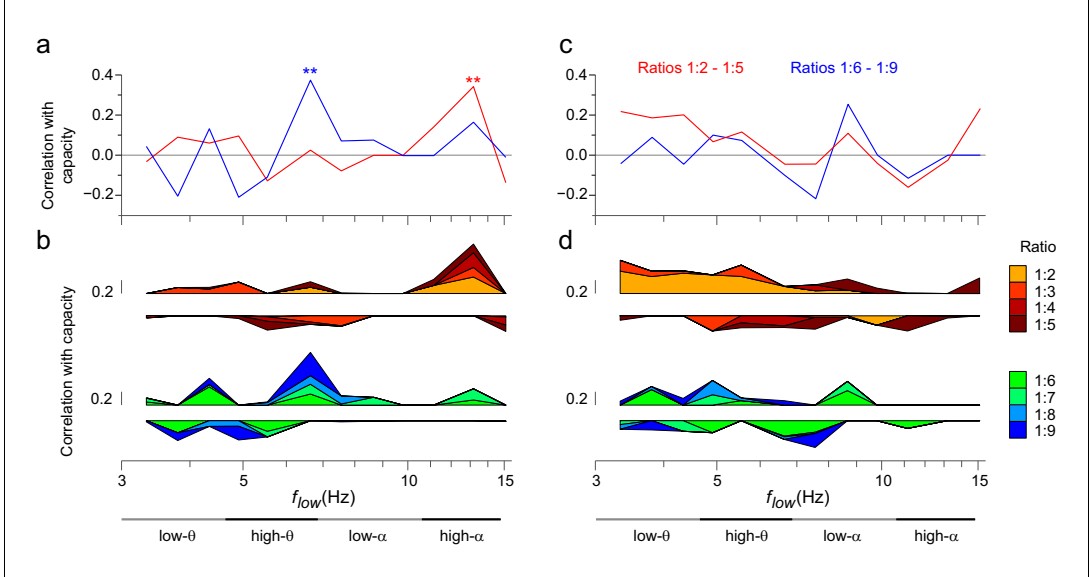

**Figure 9.** The strength of CFS predicts individual VWM capacity. (**a**) Correlation of subjects' individual CFS network strength in the Mean condition with their behavioral VWM capacity for low (1:2−1:5, red line) or high (1:6−1:9, blue line) ratios. Positive correlation is found for low ratios in the high-α band and for high ratios in the θ band (Pearson correlation, \*\*$p < 0.01$ after correction for multiple comparisons with the Benjamini-Hochberg method). (**b**) Correlation coefficients of subjects' individual CFS network strength with their VWM capacity for each ratio are shown cumulatively to illustrate the systematic correlation across ratios at high-α and θ frequencies. For the ratio colors, see legend in (**d**). Upward correlations indicate positive correlations with capacity and downward ones negative correlations. (**c**) Correlation of subjects' individual PAC network strength with their VWM capacity, visualized as in (**a**). (**d**) Correlation of subjects' individual PAC network strength with their VWM capacity for individual ratios, visualized as in (**b**).

The following source data is available for figure 9:

**Source data 1.** Statistical Table for the correlation of CFS and PAC values with the individual VWM capacity.
**Source data 2.** Statistical Table for the correlation of CFS and PAC values with the individual VWM capacity.

In line with the visualization of the most central connections (See *Figure 4b*), high-α oscillations were synchronized with γ oscillations within DA, and between DA, visual, SM, and FP systems (*Figure 6b*). As a notable feature in both of these analyses (see *Figure 6a and b*, cf. *Figure 4a and b*), was that HF oscillations in the visual system were connected with LF oscillations in DA. However, in the Load condition, high-α oscillations in SM and visual systems were synchronized with γ oscillations in visual, SM, and DA systems, and conversely γ oscillations in FP with high-α oscillations in DA, limbic, and DM systems (*Figure 6c*).

Further, in line with the visualization of strongest connections, 1:2 α:β CFS was strengthened in Mean and Load condition among SM and attentional systems (*Figure 6d and e*, cf. *Figure 4—figure supplement 1*) as well as also between DA and FP. Low-α oscillations in SM, DA, FP and DM showed suppressed synchronization with β and g oscillations in DM as well as in DA, SM and FP systems (*Figure 6f*).

Thus, these brain-system-level connectivity analyses showed that CFS connected high-α oscillations in the attentional FP and DA systems with the γ oscillations in the attentional as well visual and SM systems in the Mean condition. In the Load condition, CFS was additionally strengthened within the visual system. For the θ-consistent CFS, the connectivity pattern was more widespread and θ oscillations in many brain systems were connected to attentional networks.

## CFS connects the hubs of within-frequency (1:1) synchronized networks

If CFS mediates the integration of slow and fast 1:1 phase-synchronized networks, it should connect the most central hubs of these corresponding within-frequency (1:1) synchronized networks. To assess this hypothesis, we tested whether the most central areas of the low- ($f_{low}$) and high-frequency

($f_{high}$) CFS networks, *i.e.*, the CFS network hubs, were co-localized with the hubs of the corresponding 1:1 PS networks at $f_{low}$ and $f_{high}$. (See Materials and methods). We found that for both the high-θ (5−7 Hz) and high-α (11−14 Hz) consistent CFS networks, the node degree centrality values were correlated with the corresponding centrality values of within-frequency synchronized networks, ($p < 0.05$, corrected, *Figure 7a*, see *Figure 7—source data 1*). Hence, the low-frequency CFS hubs were significantly co-localized with the hubs of he slow 1:1 networks. The corresponding correlations of high-frequency CFS hubs with those of the fast 1:1 networks were weaker but still significant for several frequency pairs (*Figure 7a*). These findings thus support the notion that CFS connects the 1:1 networks.

We then used simplified graphs to illustrate the strongest hubs of the fast and slow 1:1 networks, their within-frequency 1:1 connectivity, and how these two networks were mutually connected by CFS. We first visualized the main hubs and their connections of 1:1 θ and α networks and the 1:2−1:3 θ:α CFS linking these networks by illustrating separately how the 1:1 θ network was connected with targets in the 1:1 α network (*Figure 7b*, left column) and how the 1:1 high-α network was connected to targets in 1:1 θ network (*Figure 7b*, right column). This showed that for LF-CFS connections, 1:1 θ and LF CFS hubs were co-localized in mFG and iFG of the LPFC and in CS (*Figure 7b*, left column) while iPG and mFG were hubs for both the 1:1 α network and HF-CFS (*Figure 7b*, right column). For 1:4−1:5 θ:β CFS and for 1:6−1:9 θ:γ CFS, shared hubs for 1:1 θ network and LF-CFS network were observed in the poCS and LPFC while shared HF-CFS and fast 1:1 hubs were observed in posterior parietal cortex (PPC) and LPFC and for or θ:γ CFS also in the visual system (*Figure 7—figure supplement 1*).

For high-α consistent CFS, we first examined the coupling of the high-α band 1:1 network with the low-γ band 1:1 network (*Figure 7c*), as well as with β-band and high-γ band networks (*Figure 7— figure supplement 2*). This revealed a very similar pattern of CFS connections than that presented in *Figure 4*. Shared key hubs in both 1:1 α and LF-CFS networks were mFS, sFG, both angular and supramarginal parts of iPG, and precentral gyrus (prCG) which were connected to their γ network targets in the visual system through CFS (*Figure 7c*, left column). Hubs shared by low-γ band and HF-CFS were SOG and inferior temporal gyrus (iTG), which were connected with their α-network targets is LPFC and PPC (*Figure 7c*, right column). Thus as proposed by our hypothesis, high-α band network included co-localized CFS and 1:1 hubs in the FP and DA attention networks while the γ-band network included key hubs in the visual system and posterior brain areas. A similar pattern of co-localized 1:1 and CFS hubs was observed also for 1:2 high-α:β (*Figure 7a* and *Figure 7—figure supplement 2a*) and for 1:6−1:9 high-α:high-γ CFS (*Figure 7a* and *Figure 7—figure supplement 2b*).

Despite both low- and high-γ networks being characterized by hubs in the visual system, they were not mutually significantly correlated in parcel centralities (*Figure 7—figure supplement 3*), which implies that these γ oscillations were relatively narrow-band and can not particularly be explained by broad-band γ activities. Overall, these data show that, as hypothesized, CFS connects 1:1 slow and fast networks through common hubs that are most prominent in PPC and LPFC as suggested also by data presented in *Figure 4*.

## Phase-amplitude coupling (PAC) and CFS are distinct phenomena

PAC has been suggested to underlie cross-frequency coupling of slow and fast oscillations and prior studies have found PAC, *e.g.*, between the θ phases and γ amplitudes during VWM (*Axmacher et al., 2010*; *Siegel et al., 2009*). To address the possibility that CFS and PAC would underlie similar cross-frequency integration, we tested whether also PAC was correlated with the VWM demands in the present data. We computed PAC between all parcels and for the same frequency pairs used in the CFS analysis. Somewhat surprisingly, as the strongest effect, this analysis revealed that θ- and low-α-band phases were coupled with α- and β-band amplitudes at a 1:2 ratio. We also found weak coupling of the γ oscillation amplitudes with the phase of θ oscillations at frequency ratios up to 1:9 (*Figure 8a*; external source data 7, *Siebenhühner et al., 2016*). Importantly, in the high-α band, where prominent CFS was observed, PAC was suppressed (*Figure 8b*). Load-dependent modulations of PAC showed a similar spectral profile (*Figure 8c–d*). These data show that PAC and CFS are clearly distinct phenomena.

## The strength of CFS predicts VWM accuracy and individual capacity

Finally, if CFS was functionally significant in the integration of neuronal processing across θ- to γ-band synchronized networks, the strength of CFS should be predictive of individual VWM task performance. We thus quantified the predictive correlation between inter-individual variability in the strength of VWM-retention period CFS and the inter-individual variability in VWM capacity across subjects. CFS network strength, $S$, was measured for each subject, VWM load, and low frequency of low (1:2–1:5) and high (1:6–1:9) CFS frequency ratios, where $S$ was the sum of phase-locking values of all significant connections in the mean condition (positive). VWM capacity $C$ was obtained separately for each subject and VWM load (See Materials and methods). We estimated the correlation (Pearson correlation test) between $S$ and $C$ and found that for low ratios the strength of high-α consistent CFS networks predicted ($r = 0.343$, $p = 0.003$) VWM capacity (*Figure 9a*, see *Figure 9—source data 1*). For the high (1:6−1:9) ratios, the individual capacity was correlated ($r = 0.376$, $p = 0.001$) with high-θ consistent CFS (see *Figure 9a*). These values remained significant when testing all low frequencies for such correlations and correcting for multiple comparisons (Benjamini-Hochberg method). We also tested whether the high-α low ratios observation was attributable only to 1:2 α:β coupling but found both the high-α:β (ratio 1:2, $r = 0.247$, $p = 0.036$) and high-α:low-γ (ratios 1:3−1:5, $r = 0.232$, $p = 0.0496$) to be significant. We also estimated (see *Figure 9b*, *Figure 9—source data 2*) the CFS-capacity correlation of all individual ratios to illustrate their systematic contribution to the correlations of their mean (see *Figure 9a*) in the high-α and θ frequency bands.

We then estimated correlations of PAC network strength with individual VWM capacity in the same manner (*Figure 9c,d*). However, no $p$ values exceeded the corrected significance threshold. These data thus indicate that cortical CFS is functionally significant in the human VWM maintenance and also further reinforce the view that CFS and PAC are functionally distinct phenomena.

## Discussion

WM is comprised of sensory storage and attentional control for manipulating the stored sensory information (*Baddeley, 1996*; *Miller and Cohen, 2001*; *Sreenivasan et al., 2014*). While the maintenance and integration of sensory information in WM is tentatively supported by β and γ phase-synchronization, θ and α phase-synchronization appears to play a role in the coordination of attentional functions. We hypothesized that concurrently synchronized assemblies in different neuronal circuits and frequency bands could be integrated into coherent VWM by cross-frequency phase synchrony (CFS).

We used M/EEG and a VWM task to investigate whether CFS would connect networks synchronized in θ-, α-, β-, and γ-frequency bands and integrate the associated neuronal processing in visual and attentional systems during VWM maintenance. We demonstrate here that human VWM maintenance is indeed characterized by large-scale networks of functionally significant CFS among cortical oscillations. We found that CFS was correlated with task demands, predicted individual VWM capacity and connected 1:1-within-frequency synchronization among task-relevant visual and fronto-parietal regions. Importantly, CFS could not be explained by changes in the signal-to-noise ratio (SNR), concurrent amplitude-amplitude correlations, or non-sinusoidal waveforms, which supports the interpretation that CFS indeed reflects dynamic cross-frequency coupling of narrow-band neuronal oscillations. The present findings thus show that CFS reflects true dynamic cross-frequency coupling of neuronal oscillations among task-relevant cortical circuits and may thus constitute a so far little acknowledged computational mechanism for integrating and coordinating neuronal processing across distinct frequency bands. Localization of the cortical sources and finding evidence for the functional significance of CFS in the present study thus extend the scarce prior observations of CFS in neuronal circuits, which have so far revealed CFS in sensor-level MEG or EEG analyses (*Hamidi et al., 2009*; *Nikulin and Brismar, 2006*; *Palva et al., 2005*; *Tass et al., 1998*), in human (*Chaieb et al., 2015*) and rat (*Belluscio et al., 2012*) hippocampal circuits, and in the neocortical (*Roopun et al., 2008*) microcircuits in rats.

## CFS connects visual and attentional systems and within-frequency synchronized networks therein

We found that in narrow θ- and high-α-frequency bands and their harmonics, CFS was positively correlated with VWM task demands. CFS hence connected the θ oscillations with those in α, β, and low-γ bands as well as the high-α with β and γ oscillations. Several studies suggest that the contents of VWM are maintained in the visual cortex (*Riggall and Postle, 2012*; *Emrich et al., 2013*; *Kravitz et al., 2013*; *Sreenivasan et al., 2014*; *Honkanen et al., 2015*), while LPFC is thought to underlie the attentional and executive regulation of the VWM maintenance (*Rowe et al., 2000*; *Miller and Cohen, 2001*; *Pessoa et al., 2002*; *Curtis and D'Esposito, 2003*; *Voytek and Knight, 2010*; *Sreenivasan et al., 2014*; *Markowitz et al., 2015*) in multiple functionally dissociable networks (*Markowitz et al., 2015*; *Lundqvist et al., 2016*). A complementary view is obtained from large-scale connectivity analyses of fMRI (*Corbetta et al., 2000*; *Corbetta and Shulman, 2002*; *Szczepanski et al., 2013*), MEG (*Spadone et al., 2015*), and intracranial EEG (iEEG) (*Daitch et al., 2013*) recordings which show that in attention tasks, functional connectivity between LPFC and PPC, *i.e.*, within the FP and DA systems, plays a fundamental functional role in attentional control and regulation. Indeed, studies both in human fMRI (*Munk et al., 2002*; *Mohr et al., 2006*) and monkey LFP (*Salazar et al., 2012*) recordings have observed strengthened activity in both frontal and parietal areas also during VWM tasks.

Hence, if CFS is a functionally significant mechanism for the integration of representational and central executive functions of VWM, it should specifically connect synchronized networks across visual and attentional systems. Moreover, as the integration in brain networks has been shown to be dependent on key brain areas that are the most central nodes, *i.e.*, the hubs in these networks (*de Pasquale et al., 2016*), CFS should connect the key hubs of the within-frequency synchronized networks among task-relevant, here attentional and visual, systems.

We found two lines of evidence for CFS playing such a role. First, the key hubs of CFS networks were largely localized to these brain systems and CFS connectivity linked these systems extensively. Both in the θ and high-α band consistent CFS networks, the low frequency oscillations had hubs predominantly in FP and to a lesser extent in DA, and significant connectivity of these hubs to faster oscillations in the visual system and DA. Conversely, the higher frequency oscillations had hubs predominantly in DA, and in the γ band also in visual cortex. Second, we found that the most central low- and high-frequency hubs of the CFS networks during VWM maintenance were anatomically co-localized with the hubs of the slow and fast, respectively, 1:1 networks and that CFS was salient among the most central nodes of the within-frequency synchronized networks between visual system and attentional networks.

The present data thus show that CFS would be well positioned to underlie the integration and regulation spectrally distributed neuronal processing between the fronto-parietal attention networks and the visual system. Thereby CFS could underlie the integration and coordination of attentional and representational functions suggested to be carried out by α and γ band synchronization, respectively (*Tallon-Baudry et al., 1998*; *Sauseng et al., 2009*; *Palva et al., 2010*, *2011*; *Bonnefond and Jensen, 2012*; *Roux et al., 2012*; *Park et al., 2014*; *Honkanen et al., 2015*).

## The roles of CFS in sensorimotor and default mode systems

CFS among high-α and β oscillations was distinct in network topography from higher-ratio CFS of high-α and γ oscillations. α:β CFS had large hubs in SM system, strong within-SM connectivity, while still having connections with both LPFC and visual system. fMRI studies have shown that during VWM, neuronal activity is also strengthened in the SM system wherein the dorsal premotor cortex (PMC) has been suggested to underlie the maintenance and manipulation of spatial features (*Munk et al., 2002*; *Mohr et al., 2006*). Thus 1:2 high-α and β CFS in SM could reflect these functions during VWM. However, α:β CFS could also be biased by the classical sensory-motor μ−rhythm (*Hari and Salmelin, 1997*) that is comprised of strongly 1:2 phase coupled a and β oscillations (*Nikulin and Brismar, 2006*). While α:γ CFS was largely distinct from α:β, also α:γ CFS involved SM hubs. Together with the prior fMRI data, this supports the notion that the SM system could be task-relevant in VWM, but further studies are needed to disentangle their functional roles.

Interestingly, both high-α- and θ-consistent CFS was found also in the DM system, including superior frontal areas of LPFC as well as inferior and superior temporal areas and the angular part of iPG.

Despite DM being often categorized as a task-negative or -irrelevant system, recent studies have shown that neuronal activity in the angular part of iPG is stronger for successfully memorized information (*Hutchinson et al., 2009*; *Lee and Kuhl, 2016*; *Spaniol et al., 2009*) while the superior frontal cortex is partly functionally similar to mFG (*Fuster, 2008*), and thus both these regions could be task-positive in the present VWM task despite belonging to DM during resting state. sTS, on the other hand, is related to multisensory processing (*Beauchamp et al., 2004*) and verbal WM functions (*Tanabe et al., 2005*) and its presence in the CSF networks might be attributable to verbal rehearsal during the memory maintenance even though the present cohort was instructed not to verbalize the task. Overall, given that CFS was predominantly observed among the visual and attentional systems, we suggest that these findings reflect task-positive roles of these brain regions rather than CFS being a phenomenon characteristic to task-negative processing during VWM.

## Suppression of CFS in task-irrelevant networks

Intriguingly, the positive CFS effects were paralleled by a suppression of CFS between δ and low-α bands with their harmonic frequency bands. The suppression involved network hubs predominantly in the SM and DM systems and at the systems-level, disengaged DA and SM from DM and FP. The DM system is widely thought to support non-task-oriented processing (*Fox et al., 2009*), which suggests that the suppression of CFS there could represent the disengagement of task-irrelevant brain regions and networks, even though some parts of DM could be task-positive in VWM (see above). Low-α consistent suppression of CFS with the concurrent increase in the high-α consistent CFS among visual and FP systems may also reflect the disengagement of neuronal processing in a task-irrelevant frequency band from that in a task-relevant band. Overall, the task-negative and -positive patterns of CFS hierarchies over the low- and high-α bands, respectively, suggest that α oscillations support both the coordination of behaviorally relevant (*Palva and Palva, 2007*; *Palva et al., 2011*; *Saalmann et al., 2012*) and the disengagement of task-irrelevant neuronal processes (*Klimesch et al., 2007*; *Jensen and Mazaheri, 2010*) as suggested previously. These results suggest that CFS can underlie both the integration as well as segregation of neuronal processing suggested to be implemented via hierarchical network interactions (*Sporns, 2013*; *Deco et al., 2015*; *Spadone et al., 2015*).

## CFS but not PAC of $\theta$ and high-α consistent oscillations predicts VWM performance

We hypothesized that CFS would regulate neuronal communication between fast and slow oscillatory networks. To obtain evidence for this, we tested whether CFS is correlated with task demands and whether it is predictive of the individual the behavioral performance. We found that CFS of θ and high-α oscillations with their harmonics in β and γ bands was strengthened above baseline levels during VWM maintenance and was positively correlated with the VWM load. As key evidence for the functional significance of this CFS, we observed that inter-individual variability of CFS network strength predicted inter-individual variability in VWM capacity. Intriguingly, the strength of high-α consistent CFS predicted individual VWM capacity at small frequency ratios while the strength of high-θ consistent CFS was correlated with individual capacity at large frequency ratios. In contrast, we found no correlations between PAC and individual capacity.

As, synchronization in θ and high-α bands is thought to support the attentional and executive functions of VWM (*Palva et al., 2010*, *2011*; *Zanto et al., 2011*; *Bonnefond and Jensen, 2012*; *Park et al., 2014*) and synchronization in γ band the contents of VWM (*Tallon-Baudry et al., 1998*; *Siegel et al., 2009*; *Salazar et al., 2012*; *Honkanen et al., 2015*), the correlation of CFS between these bands with individual capacity is well in line with the suggestion that VWM capacity is set by the cross-frequency interactions among sensory and attentional rhythms (*Lisman and Idiart, 1995*; *Lisman and Jensen, 2013*; *Roux and Uhlhaas, 2014*). However, in contrast with prior proposals based on PAC, we found only CFS to be predictive of VWM capacity.

PAC and CFS have not earlier been directly compared in cognitive tasks—the present data thus provide evidence for that these two forms of CFC are both phenomenologically and functionally distinct. The critical difference between CFS and PAC is that CFS couples the slow and fast oscillations at a temporal accuracy of the faster oscillation. CFS, but not PAC, hence can achieve consistent spike timing between these oscillations, which could reciprocally regulate neuronal communication

between the frequency bands (*Palva et al., 2005*). The present data thus show that the classical 1:1 'within-frequency' phase synchronization, which has been suggested to provide a mechanistic basis for regulating neuronal communication (*Singer, 1999*; *Fries, 2005*; *Bastos et al., 2015*; *Fries, 2015*), is complemented by cross-frequency phase-phase synchronization integrating cortical assemblies into hierarchical multi-scale networks.

## Conclusions

We found that CFS connects slow and fast cortical oscillations across the visual, FP, and DA systems during VWM retention. The strength of CFS was dynamically up- and down-regulated, correlated with VWM task demands, and predictive of individual VWM capacity, which show that CFS is a functionally significant cross-frequency interaction mechanism during VWM maintenance. CFS hierarchies can thus be dynamically recruited to meet cognitive processing demands. The present data are in line with the notions that CFS integrates neuronal processing and regulates neuronal communication among spectrally and anatomically distributed cortical networks and that CFS could thereby support the integration of attentional and representational functions of VWM.

# Materials and methods

The workflow for all steps leading to the results in *Figures 3–9* is also shown schematically in *Figure 3—figure supplement 1*, which is referenced in the following subsections by numeration. All data analyses, where not indicated otherwise, were performed on a LabVIEW-based (National Instruments, Austin, Texas) neuroinformatics platform available on request. Statistical tables for *Figures 6*, *7* and *9* are provided as source data. We further provide additional statistical data underlying *Figures 3–9* as external source data (*Siebenhühner et al., 2016*), which can be found online: dx. doi.org/10.5061/dryad.fb240.

## Task and recordings

The VWM task and MEG-EEG recordings of the data analyzed here are described in detail by Palva et al. (*Palva et al., 2010*, *2011*). In the task, 12 right-handed subjects (age 28 ± 3, mean ± SD, 4 females) memorized a Sample stimulus comprising one to six randomly located squares of different colors and after a 1 s retention period answered whether the test display was the same or different as the sample stimuli and whether they were sure about the responses. Subjects were instructed not to use verbal rehearsal during memorization. A total of ~1800 trials (~300 per memory load of 1−6 objects) were collected from each subject. Cortical activity was measured with concurrent EEG (60-channel) and MEG data (204 planar gradiometers and 102 magnetometers, Elekta Neuromag Ltd., Helsinki, Finland) recordings at 600 Hz sampling rate. Ocular signals were measured with electro-oculogram (EOG) and the behavioral thumb-twitch responses with electromyography (EMG). Only artifact-free trials with valid behavioral responses were included in the analysis. One subject was excluded from the analysis due to too large differences in head position between recording sessions. T1-weighted anatomical MRI scans for cortical surface reconstruction models were obtained at a resolution of 1x1x1 mm with a 1.5 T MRI scanner (Siemens, Germany). This study was approved by the ethical committee of Helsinki University Central Hospital and was performed according to the Declaration of Helsinki. Written informed consent was obtained from each subject prior to the experiment.

## Behavioral performance

Behavioral VWM performance was measured with Hit Rate (HR) that was given by the behavioral accuracy in each VWM load condition, *i.e.*, by the proportion of trials with correct responses of all trials that were accepted into M/EEG analyses after blink rejection.

## Data pre-processing

We used the Maxfilter software (Elekta Neuromag Ltd., Helsinki, Finland) to suppress extra-cranial noise and to co-localize the recordings in signal space (*Figure 3—figure supplement 1a*). Independent component analysis (ICA, Matlab toolbox Fieldtrip, http://fieldtrip.fcdonders.nl (*Oostenveld et al., 2011*) was used to extract signal components both for the EEG and MEG data

and to exclude components that were correlated with ocular artefacts identified by EOG or with heartbeat for which the reference signal was estimated from MEG magnetometers.

## Filtering

Preprocessed M/EEG data were filtered (*Figure 3—figure supplement 1b*) into 25 approximately logarithmically spaced frequencies from $f_{min}$ = 3 Hz to $f_{max}$ = 90 Hz using Morlet Wavelets so that the filtered time series $X_{(t,f)}$ were obtained by convolution of the original time series $x_{(t)}$ with Morlet wavelets $w(t,f)$ : $X(t,f) = x(t) \otimes w(t,f)$ for each wavelet frequency $f_{min} \leq f \leq f_{max}$, where:

$$w(t,f) = A \cdot \exp\left(\frac{-t^2}{2 \cdot \sigma_t^2}\right) \cdot \exp(2i\pi ft), \text{ where } \sigma_f = 1/2\pi\sigma_t, \ A = (\sigma_t\sqrt{\pi})^{-1/2}, \ i \text{ is the imaginary unit, and}$$

the Morlet parameter $m$ was $m = \frac{f}{\sigma_f} = 5$ (*Tallon-Baudry et al., 1996*).

## Preparation of forward and inverse operators

Source reconstruction was performed using FreeSurfer software (http://surfer.nmr.mgh.harvard.edu/) for the volumetric segmentation of the MRI data, surface reconstruction, flattening, cortical parcellation, and neuroanatomical labeling with the Destrieux atlas (*Dale et al., 1999*; *Destrieux et al., 2010*; *Fischl et al., 2002*, see *Figure 2—source data 1* for a list of parcels).

MNE software (http://www.nmr.mgh.harvard.edu/martinos/userInfo/data/sofMNE.php) was used to create three-layer boundary element head conductivity models and cortically constrained source models as well as for the M/EEG-MRI co-localization and for the preparation of the forward and inverse operators (*Figure 3—figure supplement 1c*, *Hämäläinen and Sarvas, 1989*; *Hämäläinen and Ilmoniemi, 1994*). MEG and EEG data (204 MEG planar gradiometers, 102 MEG magnetometers, and 60 EEG electrodes) were hence integrated at the inverse transform stage of data analysis (*Hämäläinen and Sarvas, 1989*; *Hämäläinen and Ilmoniemi, 1994*). The source models had dipole orientations fixed to the pial surface normals and a 7 mm inter-dipole separation throughout the cortex, which yielded models containing 7000–8000 source vertices.

## Inverse transform and vertex collapsing

A single inverse operator was used for all wavelet frequencies and was obtained by using broadband- (1–40 Hz) filtered pre-S1 baseline periods of all trials for estimating the noise covariance matrix. To reconstruct ongoing cortical phase-time series, the filtered complex single-trial M/EEG time series were inverse transformed (*Figure 3—figure supplement 1d*) to source-vertex time series that were then collapsed (*Figure 3—figure supplement 1e*) into time series of cortical parcels in a 400-parcel collection (see *Figure 2—figure supplement 2*). This parcellation was obtained by iteratively splitting the largest parcels of the Destrieux atlas along their most elongated axis using the same parcel-wise splits for all subjects (*Palva et al., 2010*; *Rouhinen et al., 2013*; *Honkanen et al., 2015*). Using neuroanatomical labeling as the anatomical 'coordinate system' made inter-subject morphing in group-level analyses unnecessary, which would have compromised the individual anatomical accuracy. Collapsing the source time-series into time-series of cortical parcels was performed with optimized collapse operators where only the source vertices with greatest source reconstruction accuracy were used (*Korhonen et al., 2014*). For creating the collapse operators, we used forward and inverse modeling of simulated data to assess the source reconstruction accuracy for each source vertex and cortical parcel separately for each subject. In these simulations, Morlet-filtered white noise was created independently for each cortical parcel and these time series together were forward modeled to simulate the M/EEG recording of ongoing brain activity with known source dynamics. Inverse modeling and parcel time series collapsing, carried out identically to the inverse modeling of real data, was then applied to the simulated M/EEG data to reconstruct the spatial correlation patterns attributable to signal mixing in M/EEG data acquisition and source reconstruction. Comparison of the original parcel time series with the forward-inverse modeled source-vertex time series was then used to identify for each parcel the collection source vertices that most accurately reconstruct the original parcel time series (*Korhonen et al., 2014*).

## Removal of low-fidelity vertices

In phase synchrony analyses of collapsed parcel time series data, one major confounding factor is spurious edges resulting from signal mixing between neighbouring brain regions in data acquisition

and source reconstruction (*Palva and Palva, 2012*). We assessed the reliability of data based on phase correlations between real and simulated data. *Parcel fidelity (f)* was defined as the phase correlation between the original true parcel time series and the forward-inverse modeled parcel time series (*Korhonen et al., 2014*). *Parcel cross-talk (c)*, conversely, was taken as the phase correlation between the forward-inverse modeled parcel time series with the original true time series of all other parcels. For each parcel, we calculated *c* as the average of *f* values with all other parcels. Fidelity hence measures the parcel reconstruction accuracy and parcel cross-talk summarizes the degree of mixing a given parcel has with other parcels. *Parcel spread (s)* for each parcel *p* was calculated as the mean *f* of all other parcels with the original time series of parcel *p* and hence reveals parcels generating signals that are spuriously picked up by many other parcels. To decrease the probability of spurious synchronization and anatomically misplaced connections, for both statistical analyses and graph visualization in this study, we removed parcels with $\frac{f}{c \cdot s} < 3000$ (a.u.), so that ~10% of the cortical surface was excluded. These parcels were located mostly deep and/or inferior sources, which generate the least detectable signals in M/EEG and hence are most likely to incorrectly reflect signals generated elsewhere.

## 1:1 and CF synchrony analysis

We estimated 1:1 phase-synchrony within frequencies and m:n phase-synchrony between frequencies from source-reconstructed phase-time series of M/EEG data (*Figure 3—figure supplement 1f*). The filtered time series $X(t,f)$ are complex and can be expressed as amplitude and a phase time series $A_x(t,f)$ and $\theta_x(t,f)$, respectively, so that

$$X(t,f) = A_x(t,f) \cdot exp[i \cdot \theta(t,f)].$$

We estimated $n:m$ cross-frequency phase synchrony (CFS) and 1:1 within-frequency phase synchrony for each pair of cortical parcels $p$ and $q$ and between pairs of low and high frequencies $f_{high}$ and $f_{low}$. We used the phase-locking value (PLV) to compute synchronization so that

$$PLV_{p,q,n:m,f_{low},f_{high}} = \frac{1}{N} \left| \sum_{r,t} \exp\left[ i \cdot \left( m \cdot \theta_p(r,t,f_{low}) - n \cdot \theta_q(r,t,f_{high}) \right) \right] \right|$$

where $i$ is the imaginary unit, the integers $n$ and $m$ define the frequency ratio so that $n \cdot f_{high} = m \cdot f_{low}$, $N = N_r \cdot N_t$, where $N_r$ is the number of trials $r$ and $N_t$ is the number of samples $t$ within a time window (*Palva et al., 2005*).

For CFS we used n = 1 and m ∊ {2,3,4,5,6,7,8,9} while for 1:1 within-frequency phase synchrony m = 1.

Phase-amplitude coupling (PAC) was also quantified for the same frequency pairs as CFS. We estimated PAC by computing the 1:1 PLV between the phase of the slow oscillation and the phase of the amplitude envelope of the fast oscillation (*Vanhatalo et al., 2004*). The phase of the amplitude envelope was estimated by filtering it at $f_{low}$. PAC was thus defined as:

$$PAC_{p,q,f_{low},f_{high}} = \frac{1}{N} \left| \sum_{r,t} \exp\left[ i \cdot \left( \theta_p(r,t,f_{low}) - \theta_q^E(r,t,f_{low},f_{high}) \right) \right] \right|$$

where $\theta^E(t,f_{low},f_{high})$ is the phase of the filtered amplitude envelope time series $E(t,f_{low},f_{hight})$ that was obtained by filtering $A(t,f_{high})$ with the Morlet wavelet $w(t,f_{low})$:

$$E(t,f_{low},f_{high}) = A(t,f_{high}) \otimes w(t,f_{low}).$$

Cross-frequency amplitude correlations were estimated for the same frequency pairs as the CFS by using the Pearson correlation coefficient *CC* of the amplitude time series *A(t)* so that

$$CC_{p,q,f_{low},f_{high}} = \frac{1}{N} \sum_{r,t} \left( Z_p(r,t,f_{low}) \cdot Z_q(r,t,f_{high}) \right)$$

where $(f,r,t) = \frac{A(f,r,t) - \mu_{f,r,t}}{\sigma_{f,r,t}}$, $\mu_{f,r,t} = \frac{1}{N} \sum_{f,r,t} A(f,r,t)$ and $\sigma_{f,r,t} = \sqrt{\frac{\sum_{f,r,t} \left( A(f,r,t) - \mu_{f,r,t} \right)}{N}}$.

## Single-trial analysis

To assess 1:1 synchrony and 1:4 CFS at the single-trial level (see *Figure 2*), we computed the 1:1 and 1:4 PLV for a single trial between parcels observed to exhibit clear 1:4 CFS in the group analyses (see below and *Figure 4*) in sliding 300 ms wide time windows at 50 ms distance. While 1:1 PLV estimates may be biased by residual linear mixing after source reconstruction, neither the long-range couplings nor the strengthening of 1:1 PLVs during the retention period can be attributed to linear mixing. To assess the statistical significance, we computed the 99th% -ile of the PLV distribution of 1000 realizations of surrogate data. The surrogate PLVs were assessed identically to those of real data and for the same sliding time windows, but so that one of the time series was shifted randomly by 0−300 ms. This method of surrogate generation is specific for measuring the significance of inter-areal correlations because all local features that bias the PLV estimates, such as temporal auto-correlations, are preserved.

## Group-level statistics

We then computed m:n CFS, PAC, and 1:1 phase-synchronization across trials separately for all subjects ($N$ = 12) between all cortical parcels, combinations frequency pairs of $f_{low}$ and $f_{high}$, and for four time-windows; two before stimulus onset and for early (400–700 ms from Sample stimuli) and late (700–1000 ms from Sample stimuli) retention periods and for all memory loads $l$ ($N$ = 6).

Since PLV is sensitive to the number of trials, we equalized the number of trials between conditions (memory loads) within each subject to the lowest number of trials among conditions. Prior to statistics, the CFS adjacency matrices were collapsed (*Figure 3—figure supplement 1g*) to a coarser parcellation, the Destrieux atlas (*Destrieux et al., 2010*) of 148 parcels (see *Figure 2—figure supplement 2*).

We then estimated (*Figure 3—figure supplement 1h*) the statistical significance across the entire subject population of each pairwise interaction for each of the 1:$m$ ratios. We first identified interactions that were significantly stronger or weaker in the retention period time windows than in the first pre-stimulus baseline by using a two-sided t-test with a statistical significance level p<0.05, and corrected for multiple comparisons for data averaged across the six memory loads (Mean condition). The effect sizes (Mean difference PLV$_{ret}$–PLV$_{BL}$ divided by the standard deviation of the difference) and $p$-values for all frequencies and ratios are provided in external source data 1 (*Siebenhühner et al., 2016*). To correct for multiple comparisons, we removed as many of the least significant edges as predicted by the $p$-value (False discovery rate correction). For the effect of memory load (Load condition), we identified interactions whose strength was positively or negatively correlated with the memory load with a Spearman rank correlation test (p<0.05; FDR corrected). In the Load condition, the effect size is given by the correlation coefficients that together with the $p$ values are provided in external source data 1 (*Siebenhühner et al., 2016*).

We then used graph theoretical notation (*Bullmore and Sporns, 2009*) to characterize the statistical CFS adjacency matrices at the graph level. Here, brain areas are the nodes, and connections of synchrony, the edges. We used connection density, $K$, that was defined to indicate the proportion of statistically significant inter-areal interactions from all possible interactions to indicate the extent of statistically significant synchronization separately for each frequency ratios of CFS (*Figure 3*), CF amplitude-amplitude correlations (*Figure 3—figure supplement 7*) and PAC (*Figure 8*). The effect sizes and $p$ values for CFS, amplitude-amplitude correlations and PAC are provided in external source data 1, 2 and 7, respectively (*Siebenhühner et al., 2016*).

To compare the overlap between CFS and CF amplitude-amplitude coupling, we then computed the fraction of edges that were found significant in both CFS and CF amplitude-amplitude correlations to those that were significant in CFS only (*Figure 3—figure supplement 8*).

## Estimation of consistency across ratios

To quantify how consistently the connection density $K$ of CFS was increased or decreased across the 1:$m$ frequency ratios for each $f_{low} < 18$Hz, we introduced (*Figure 3—figure supplement 1i*) a consistency measure $C$:

$$C_{l,f_{low}} = \bar{K}_{l,f_{low}} \cdot exp\left( -\frac{\sum_{m}^{N} |K_{m,l,f_{low}} - \bar{K}_{l,f_{low}}|}{N \cdot \bar{K}_{l,f_{low}}} \right)$$

where $l$ denotes the memory load or test condition (Mean or Load), $K_{m,i,f_{low}}$ is the connection density for $\mathrm{CFS}_{low:high}$ at load $l$ and $\bar{K}$ is the mean value for all $K$, $N$ is the number of ratios (here 8). The $C$ values were considered significant if they exceeded 95% -ile confidence limits estimated from 2000 realizations of surrogate data that were constructed by shuffling the $K$ values over frequencies. The same procedures described in this paragraph were also used for estimation of $K$ and $C$ for amplitude-amplitude (CC) and phase-amplitude (PAC) couplings.

## Construction of summary graphs across ratios

We then performed (*Figure 3—figure supplement 1j*) graph visualization for CFS graphs in two stages. In the first stage, we created graphs that summarize the connections shared among harmonically consistent networks across frequency ratios. These summary graphs are provided in external source data 3–6 (*Siebenhühner et al., 2016*). In the second stage, the graphs were visualized so that the most central edges were selected and hierarchically clustered to attenuate the effect of spurious edges.

In the first stage, in order to identify CFS edges that were shared across ratios, edges from all ratios of interest were clustered into hyperedges where each hyperedge could contain edges from the adjacency matrices of multiple ratios. This edge clustering was performed so that an edge-edge adjacency matrix was created and the adjacency was defined to be the product of corresponding maximum parcel cross-talk $c$ (see previous paragraph). Thus each element in the edge-edge adjacency matrix indicates the proximity of any two observed edges in terms of linear mixing attributable to data acquisition and inverse modeling on the group level. Here high adjacency indicates a high probability that these edges actually represent the same underlying cortico-cortical interaction. Hierarchical clustering of the edge-edge adjacency matrix thus groups all observed edges into a set of hyperedges that are mutually dissociable in the MEG linear mixing space. To achieve a representation of connectivity shared by multiple frequency ratios, we used permutation statistics to test for each hyperedge whether it could be significantly (p<0.05, uncorrected) attributable to edges from any single ratio. Then those hyperedges for which this was the case were excluded from further visualization. The remaining hyperedges were then 'flattened' into a regular adjacency matrix by summing the shared edges across ratios.

## Bundling of edges into hyperedges

In the second stage (*Figure 3—figure supplement 1k*), in order to visualize these graphs for the Mean and Load conditions we picked the 500 and 250 most central edges, respectively, i.e., edges that were connected to the most central low- or high-frequency hubs. For $f_{low}$, we ranked the edges by the low-frequency degree centrality of the vertices to which their low-frequency end was connected. Conversely for $f_{high}$, we ranked edges by the high-frequency degree centrality of the vertices that their high-frequency end was connected to.

We then re-used the edge-adjacency based hierarchical clustering described above in order to bundle the edges into clusters that are likely to each represent a single underlying true interaction. A bundle *per se* indicates the residual uncertainty about the anatomical localization of the interaction. As large bundles are much more likely to reflect major true neuronal interactions than small bundles, we excluded bundles of less than 4 edges and discarded edges with low centrality (<50% ) within a bundle.

We created low-and high-frequency graphs, *i.e.*, graphs that showed the most central LF or HF hubs and their connections to other vertices (parcels), respectively. Circles were used to denote the vertices, *i.e.*, parcels, so that their size was proportional to their degree in the graph. The vertices were coloured by the functional brain system (*Yeo et al., 2011*) they belonged to. As many vertices would be both at the high and low-frequency 'end' of connections within a graph, we chose to indicate their fractions of LF and HF degree by the light and darker shaded sectors. For example, a vertex with 3 connections where it was at the LF 'end' and one where it was at the HF 'end' would be divided into 75% area of lighter and 25% area of darker shade. CFS connections were indicated by lines where dashed ends pointed towards the high-frequency vertices in LF graphs and towards low-frequency vertices in HF graphs. The neuroanatomical spread of the ends of each bundle illustrates the level of spatial smearing in M/EEG source reconstruction used here. The graphs were overlaid on flattened maps of the complete cortical surfaces of left and right hemispheres. These maps

indicate the cortical gyration so that sulci are darker than gyri. Color in flattened brains identifies the 7 brain systems of the Yeo parcellations (*Yeo et al., 2011*) and the thin white or grey lines the 148 parcels of the Destrieux parcellation (*Destrieux et al., 2010*). These graphs are shown in *Figures 4*, *5* and their supplements.

## Assessing interactions within and between functional brain systems

To estimate the interactions between functional subsystems, we morphed (*Figure 3—figure supplement 1l*) the ratio-collapsed graphs graphs from 148x148 parcel interaction matrices into 7x7 subsystem interaction matrices, based on the Yeo 2011 parcellation scheme (*Yeo et al., 2011*) and computed for each element the connection density *K*. We then computed 5000 randomized versions of the same 148x148 matrix, keeping the number of edges constant. *K* values of the true subsystem interaction matrix were reported as significant (see *Figure 6*, *Figure 6—source data 1*), if they exceeded the 95th percentile of *K* values in randomized graphs.

## Correlation between networks of CFS and 1:1 phase synchrony

To assess whether the CFS connected 1:1 synchronized networks, we obtained (*Figure 3—figure supplement 1*) Mean condition 1:1 networks for all analyzed frequencies with the statistical approach identical to what was used for CFS. With these networks, we tested whether the vertex centralities (represented by vertex degree) of the low-frequency ($f_{low}$) CFS network vertices were correlated with centrality values for the vertices of the corresponding slow 1:1 networks. Conversely, we tested whether the vertices of the high-frequency ($f_{high}$) part of the CFS networks were correlated with the vertices of fast 1:1 networks. Finding such correlations would indicate that CFS connects the major nodes of spectrally distinct within-frequency synchronized networks. We used the vertex degree to quantify the centrality and Pearson correlation to quantify the correlations. The Pearson correlation *p* values were corrected for multiple comparisons with the Benjamini-Hochberg FDR method (*Figure 7a*, *Figure 7—source data 1*). For visualization (*Figure 7b–c* and *Figure 7—figure supplements 1* and *2*), we used standard force-directed graph drawing.

## Degree correlation between frequencies

To assess the parcel centrality similarity between narrow-band 1:1 networks, we computed the vertex degree for each parcel in the Mean condition (positive tail) networks of all 25 narrow-band frequencies. We then evaluated, for each pair of frequencies, the correlation of vertex degrees across all parcels with the Pearson's correlation coefficient. The results are presented in *Figure 7—figure supplement 3*.

## Correlation of CFS and PAC with the behavioral performance

To assess whether CFS and PAC were predictive of the subjects' behavioral VWM performance, we estimated (*Figure 3—figure supplement 1n*) the correlation between the network strength of CFS networks and individual capacity for each memory load. Individual CFS strength was obtained as the sum of individual connection strengths across connections significant in the positive tail of Mean condition group statistics. Individual VWM capacity was defined for each VWM load as $C(l) = HR(l) \cdot l$, where $HR(l)$ is the hit rate at load $l$.

In detail, the interaction matrices for the Mean condition were used to create binary masks $M$. If the interaction at the 1:*m*-ratio *m*, at low frequency $f_{low}$, in time window *t* was found significant between parcels *p* and *q* in the group-level analysis, $M(p, q, m, t, f_{low})$ was set to 1, otherwise to 0. For each subject, we multiplied CFS and PAC adjacency matrices with these masks and then summed over all parcel pairs. Subjects' individual network strength $S$ for each *m*, memory load *l*, and frequency $f_{low}$ was thus calculated as:

$$S\left(l, m, f_{low}\right) = \sum_p \sum_{q \neq p} \sum_{t=3,4} M(p, q, m, t, f_{low}) PLV(p, q, l, m, t, f_{low})$$

Prior to correlation analyses these network strength and VWM capacity values were detrended and normalized. For each *m* and $f_{low}$, the modified individual strength $S'$ of each subject *s* was obtained by zero-meaning with subtraction of the individual mean *S* across loads, $\bar{S}_L(s) = \frac{1}{N_l} \sum_l S(s, l)$, detrending with the population mean *S* for each load $\bar{S}_{pop}(l) = \frac{1}{N_s} \sum_s (S(s, l) - \bar{S}_L(s))$, and normalizing by dividing

with the absolute population mean $\bar{S}_{abs} = \frac{1}{N_s} \sum_s \sum_l |S(s,l) - \bar{S}_L(s)|$, so that: $S'(l,s) = \frac{1}{\bar{S}_{abs}} (S(l,s) - \bar{S}_L(s) - \bar{S}_{pop}(l))$.

Identical detrending and normalization was applied to the VWM capacity values, *i.e.*, for obtaining $C'(s, l)$ from $C(s\ l)$. We then averaged $S'$ separately across low (1:2−1:5) and high (1:6−1:9) CFS ratios and then computed for each $f_{low}$ the correlation of $S'$ and $C'$ across subjects and VWM loads with the Pearson correlation test (see *Figure 9a and c*, *Figure 9—source data 1*). Correction for multiple comparisons across all $f_{low}$ ($n = 12$) was performed using the Benjamini-Hochberg method. For dissecting *post hoc* the role of 1:2 CFS (see Results), we further also evaluated these correlations for the ratios 1:2 and 1:3−1:5 for $f_{low} = 13$ Hz. For visualization without statistics (see *Figure 9b and d*, *Figure 9—source data 2*), we also computed correlation coefficients separately for individual ratios.

### Leave-one-out and effect size analyses

We performed leave-one-out and effect-size analyses to confirm that the findings of increased CFS (Mean condition) and of CFS positively correlated with VWM Load (Load condition) were not affected by the small sample size of our cohort. In the leave-one-out analyses, we created 12 data-sets of 11 subjects, where in each one a different subject had been excluded and repeated the statistical analyses underlying *Figure 2a and c*. We then retained those edges that were found significant in the original analyses and had a p-value<0.05 in each leave-one-out dataset. These results are presented in *Figure 3—figure supplement 2a* and d for Mean and Load conditions, respectively. We then estimated that with 12 subjects in our cohort, to get alpha level 0.05 with statistical power of 0.8, our data needed to have an effect size of minimum of 0.9 for the Mean condition and 0.35 for the Load condition. In the effect size analyses, we thus retained those edges which were found significant in the original analyses and which had an effect size > 0.9 (Mean condition) or a correlation coefficient > 0.35 (Load condition). (*Figure 3—figure supplement 2b and e*). Finally, we show the combined results from these masks in *Figure 3—figure supplement 2c and f*, where only the edges significant in both the leave-one-out and effect-size analyses are retained. The same analysis was also done for PAC, with the results shown is *Figure 8—figure supplement 1*.

### Comparison of observed changes in PLV with changes predicted from changes in SNR or oscillation amplitudes

Changes in phase synchrony may be partially caused by changes in oscillation amplitude. In particular, an increase in signal-to-noise ratio (SNR) can improve phase estimation and therefore also phase-locking values (PLV). However, we showed in our earlier analysis of the same dataset that changes in SNR could only explain a small fraction of the observed changes in PLV (*Palva et al., 2010*). We here extend the approach to the case of cross-frequency phase synchrony.

We simulated four time series *X*, *Y*, $N_x$ and $N_y$ as white noise with $10^6$ samples in the range of -1 to 1 that was filtered with a Morlet wavelet at $f_1$ for *X* and $N_x$ and at $f_2$ for *Y* and $N_y$, where $f_2 = n \cdot f_1$, obtaining complex time series that can be written in the form $X = A_x \cdot exp(i \cdot \varphi_x)$ and analogously.

We then obtained coupled time series as follows: $X' = X + c \cdot A_y \cdot exp(\frac{i}{n}\varphi_y)$ and $Y' = Y + c \cdot A_x \cdot exp(in\varphi_x)$, where $c$ denotes the coupling factor. *X'*, *Y'*, $N_x$ and $N_y$ were then normalized as in $X'_n = X'/|X'|$ and so on, and the noise added to the coupled time series series: $X'' = X'_n + s_x \cdot N_{x,n}$ and $Y'' = Y'_n + s_y \cdot N_{y,n}$, where $s = 1/SNR$.

Then, we estimated CFS, using PLV, between X'' and Y' for a range of values each for $s_x$ and $s_y$, which were spaced so that at $SNR_{i1} = 1.3 \cdot SNR_i$. We estimated the apparent SNR (*aSNR*) by comparing the mean amplitude levels for experimental data across object loads and empty-room recordings that were obtained before or after each recording session. Empty-room data were inverse modeled with the inverse operator derived from the experimental data onto 400 parcels, and we defined the *aSNR* as follows:

$$aSNR = \frac{1}{N_s \cdot N_p} \sum_s \sum_p \frac{A_{exp,s,p} - A_{ER,s,p}}{A_{ER,s,p}}$$

where $A_{exp,s,p}$ denotes the mean amplitude for experimental data and $A_{ER,s,p}$ the mean amplitude for

empty room recordingsfor subject $s$ (here $N_s = 6$) and parcel $p$. The values for aSNR are shown in *Figure 3—figure supplement 3*.

The (non-linear) function linking aSNR and SNR was estimated numerically. We created white noise time series, noise and signal plus noise time series as above and obtained $aSNR = \frac{A_{S+N} - A_N}{A_N}$, where $A_{S+N}$is the mean amplitude of the signal plus noise time series and $A_N$ the mean amplitude of the noise time series. By interpolating this function, we estimated SNR values for experimental data from aSNR values.

We used these results to predict how large changes in PLV would be caused by changes in amplitude in the low and high frequency band for selected frequency ratios (those that showed the largest K values). For each of these, we identified the 200 parcel pairs with the largest observed change in PLV in order to focus the analysis on the strongest effects in this dataset. If PLV were significantly biased by SNR changes, this interaction should be most salient in the strongest PLV findings.

To obtain the changes of PLV in experimental data in the Mean condition, we computed for each parcel pair $j$, $k$, the difference of the mean PLV between initial and modulated condition. For the Mean effect, we computed the difference of the mean over subjects and object loads $l$ in the retention period with that in the baseline window:

$$\Delta PLV_{exp,j,k,f_1,f_2} = \frac{1}{N_s \cdot N_l} \sum_s \sum_l PLV_{s,j,k,f_1,f_2,ret} - PLV_{s,j,k,f_1,f_2,BL}$$

For the Load effect, we computed instead the difference of the 6 object and 1 object conditions in the retention period:

$$\Delta PLV_{exp,j,k,f_1,f_2} = \frac{1}{N_s} \sum_s PLV_{s,j,k,f_1,f_2,ret}(l=6) - PLV_{s,j,k,f_1,f_2,ret}(l=1)$$

We further estimated the SNR for these conditions and parcel pairs from the aSNR and used the SNR levels and PLV for the initial condition to estimate the coupling factor $c$ from the simulated time series, for which:

$$PLV_{exp,init}(f_1,f_2) = PLV_{sim,f_1,f_2}\left(c_{in}, SNR_{1,init}, SNR_{2,init}\right)$$

We then estimated the predicted PLV from the simulated time series with the SNR levels in the modulated condition:

$$PLV_{pred}(f_1,f_2) = PLV_{sim,f_1,f_2}\left(c_{in}, SNR_{1,mod}, SNR_{2,mod}\right)$$

and thus obtained the change in PLV that could be expected from amplitude changes alone:

$$\Delta PLV_{pred} = PLV_{pred} - PLV_{exp,init}$$

The results for the Mean and Load conditions are shown in *Figure 3—figure supplements 4* and *5*, respectively.

To assess whether PLV and amplitude effects were correlated directly, we calculated the Mean- and Load-condition effects identically to what was done in the main analyses, for each ratio, frequency and parcel pair. In the Mean condition, we computed the average PLV difference between retention period and baseline time windows over subjects and conditions. In the load condition, we correlated the PLV values in the retention period with the object load across subjects. We then morphed these data from the 400-parcel to the coarser 148-parcel Destrieux parcellation. In the same manner, we computed for each frequency and ratio the Mean and Load effects for the measured parcel amplitudes, averaged these amplitude effects for each parcel pair, and morphed to the Destrieux parcellation. For both Mean and Load condition, we then computed the correlation coefficient $r$ between the PLV and amplitude effect for all parcel pairs that were found to have a significant PLV effect in the corresponding group analysis. We compared these $r$ against 1000 surrogate datasets where the amplitude values were randomly shifted across edge pairs. We considered the PLV-effect vs. amplitude-effect correlation significant if the $r$ was in the highest or lowest 2.5%-ile of the surrogate values. No multiple comparison corrections were applied across the studied frequency-ratio pairs. In *Figure 3—figure supplement 6*, we illustrate the fraction of variance in PLV effects

explained by the amplitude effects with $r^2$ values for those ratios and frequencies that had a significant correlation between the PLV and amplitude effects and initially had K > 0.1% in the group analysis.

## Additional information

### Funding

| Funder | Grant reference number | Author |
|---|---|---|
| Brain and Mind doctoral program | | Felix Siebenhühner |
| Suomen Akatemia | SA253130 | J Matias Palva |
| Suomen Akatemia | SA281414 | J Matias Palva |
| Suomen Akatemia | SA256472 | J Matias Palva |
| Suomen Akatemia | SA1126927 | Satu Palva |
| Helsinki University Research Grants | 788/51/2010 | Satu Palva |
| Suomen Akatemia | SA266402 | Satu Palva |

The funders had no role in study design, data collection and interpretation, or the decision to submit the work for publication.

### Author contributions

FS, SHW, Analysis and interpretation of data, Drafting or revising the article; JMP, Conception and design, Analysis and interpretation of data, Drafting or revising the article; SP, Conception and design, Acquisition of data, Analysis and interpretation of data, Drafting or revising the article

### Author ORCIDs

Satu Palva, http://orcid.org/0000-0001-9496-7391

### Ethics

Human subjects: This study was approved by the ethical committee of Helsinki University Central hospital and was performed according to the Declaration of Helsinki. Written informed consent was obtained from each subject prior to the experiment.

## Additional files

### Major datasets

The following dataset was generated:

| Author(s) | Year | Dataset title | Dataset URL | Database, license, and accessibility information |
|---|---|---|---|---|
| Siebenhühner F, Wang SH, Palva JM, Palva S | 2016 | Data from: Cross-frequency synchronization connects networks of fast and slow oscillations during visual working memory maintenance | http://dx.doi.org/10.5061/dryad.fb240 | Available at Dryad Digital Repository under a CC0 Public Domain Dedication |

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
