## [Decision Letter]

Thank you for submitting your article "Cross-frequency synchronization connects networks of fast and slow oscillations during visual working memory maintenance" for consideration by *eLife*. Your article has been reviewed by three peer reviewers, and the evaluation has been overseen by a Reviewing Editor and David Van Essen as the Senior Editor.

The reviewers have discussed the reviews with one another and the Reviewing Editor has drafted this decision to help you prepare a revised submission.

All the reviewers recognize the importance of your work in which you show the importance of cross-frequency synchronization (CFS) as a mechanism for linking neuronal populations that have different oscillation patterns. They are also quite enthused about the sophisticated methodology and analytic strategies. The relationship with performance (load) in the working memory task is also considered a strength.

However, the reviewers have raised significant issues that need to be addressed in a revision. The most important issue is the statistical robustness of the results given the small sample size and multiple comparisons run across many conditions. There are also questions related to the robustness of cross-frequency in relation to the amplitude of the signal, and some issues related to the definition of CFS networks. Secondly, the relevance and advance provided by the results is not clear.

Thirdly, the readability of the paper should be improved. At the same time it was felt that some of the results were presented in a highly summarized form that were difficult to evaluate.

Major:

Robustness of results

1) The sample (N=12) is very small for a correlation analysis. This is compounded by the number of correlations computed. No scatterplots are shown to illustrate whether the findings may have been driven by outliers. It's also not very clear in Figure 8 which correlations are significant, as the asterisks seem to be on top of all four bars. There are more comparisons than subjects.

2) Potential confounds due to changes in signal amplitude are not sufficiently addressed. The authors correctly point out potential confounding of cross-frequency synchrony (CFS) by changes in signal-to-noise ratio due to task-induced changes in oscillations amplitudes. However, the reported control analysis of amplitude-amplitude coupling does not address this confound, because it tests for changes in power co-variance, but not for changes in the mean power between baseline and memory intervals. However, any change in power, and thus SNR, of merely one of two analyzed nodes is expected to yield a spurious change in measured CFS even without any true change in underlying CFS. Critically, one may observe spurious decreases and increases depending on the relative strength of CFS and amplitude changes for signal and noise components. In previous analyses of the same data, the authors reported amplitude changes during visual working memory (Palva et al., J Neurosci 2011) that were load-dependent and performance-predictive just like the CFS effects reported here. It should be ruled out that the present CFS results reflect spurious effects driven by these amplitude changes. E.g., the authors should directly compare CFS and amplitude effects and may stratify the data for potential amplitude changes before quantifying CFS.

3) The interpretation of network results seems at odds with the shown data at several points. The description of CFS networks (Figure 4; Results section) stresses the claimed link between executive (FP, DA) and visual networks. However, this does not become clear from Figure 4, Figure 5 and 6. E.g., it is stated that FP high-α connects to visual γ, or more generally, that theta and high-α consistent CFS connects the visual system with FP and DA networks. This is neither readily apparent from Figure 4 nor from Figure 6. In fact, in contrast to what is stressed in the text, there seem to be hardly any visual hubs in Figure 4. At the same time, the sensorimotor network, which the authors refer to as putatively task-irrelevant, consistently shows up in CFS networks (Figure 6), but this is not discussed. Similarly, the authors suggest CFS suppression of task-irrelevant SM and DM networks, but do not mention that CFS was also suppressed in putatively task-relevant DA and FP networks (Figure 6, 4th matrix). The description of networks should be clarified and inconsistencies with the hypothesis should be pointed out. It may also help to simplify or split the network display of Figure 4 and Figure 5, which convey a lot of information, but are hard to parse.

4) The evidence supporting a link between CFS and 1:1 networks is weak (Figure 3; Results section). The reported correlations between networks seem low. Furthermore, these correlations may be driven by global gradients rather than by a match of distinct network hubs. I suggest showing CFS and 1:1 networks (degree and/or betweenness) to substantiate the suggested match.

5) Why were high-α consistent networks only investigated for ratios 1:3-1:9, but not for 1:2? 1:2 CFS showed mean and load effects (Figure 2). This seems like an arbitrary sub-selection.

6) Inferential statistics (with p-values) seem to be missing for the claimed increases and decreases of individual-frequency CFS during VWM (Figure 2).

7) No spectral graphs are shown, or comparisons of topographical maps with source reconstructions.

8) In Figure 2, significant interactions span many n:m ratios. This seems to be a robust observation, as subsequent figures collapse over many frequency pairs from 1:2 or 3:9. I think this is what they refer to as harmonic consistency. But the interpretation is not very clear; I think it means that the coupling is actually with fast broadband fluctuations and they are only measuring these fluctuations at a limited number of sampled frequency points. If these reflect rhythmic interactions, the authors would need to show that there are separated spectral peaks at each integer multiple. I wouldn't consider low frequencies coupled to broadband fluctuations a "confound" (I guess the authors wouldn't either), but the interpretation of phase-phase oscillatory synchronization is quite different from the interpretation of phase-broadband coupling.

Relevance and Advance

This is a very rich and complicated set of results linking synchronization across different temporal scales to visual working memory performance. There are a lot of results and it is not clear how they are all related. The take-home message seems to be that the brain is complicated and produces myriad dynamics. Of course I don't doubt such a conclusion, but it is difficult to understand the specific contribution to the literature. In other words, what have we learned here?

Readability

The findings are significant and important, but I'm concerned about the overall readability. I consider myself fairly fluent in these domains (CFC, graph theory) but it still took me quite some time to wrap my head around the results enough to put them into a mechanistic and cognitive context. I strongly believe that a clear schematic figure, up front, would help guide readers through the rest of the paper.

While I recognize that the current Figure 1 attempts to do this, there are certain design features that hinder it. For example, upon my first viewing, the fact that the colors used to differentiate cortical regions and connection in 1a (blue, purple, green, red, yellow) are the same colors chosen for the squares when illustrating a VWM trial suggests a link between the two when, in fact, they are unrelated. To be honest, my first thought was that there was so spatial decoding element to the paper.

[Editors' note: further revisions were requested prior to acceptance, as described below.]

Thank you for resubmitting your work entitled "Cross-frequency synchronization connects networks of fast and slow oscillations during visual working memory maintenance" for further consideration at *eLife*. Your revised article has been favorably evaluated by David Van Essen (Senior editor), Maurizio Corbetta (Reviewing editor), and three reviewers.

All reviewers found that the revised manuscript addresses most of their concerns.

I would ask you to consider Reviewer 2 comments on the SNR and description of Figure 7.

We hope that these issues can be dealt with expeditiously.

Reviewer #1:

In my opinion, the authors have done a fine job at addressing the Reviewers' concerns. In particular I appreciate the new cross-validation analyses.

Their new Hypothesis Figure 1 is very helpful, and is a useful addition overall.

I am in the awkward, but pleasant, position of not having any major questions or concerns.

Reviewer #2:

The authors have extensively revised the manuscript and included several new analyses. This has substantially strengthened the manuscript and most of my concerns have been addressed. Overall, this is a very good paper reporting several intriguing and novel results. However, two points remain:

1) I am concerned that the SNR confound analysis is based on the invalid assumption that only the measurement noise (assessed with an empty room measurement) constitutes noise in the present context of a potential SNR confound. In contrast, for this confound, also any neuronal signals not of interest (and likely without CFS) should be considered noise. In particular, this noise entails broad band (1/f) activity, which constitutes a major part of the measured neuronal signal. For any frequency band, the measured neuronal signal does not only reflect oscillatory and potentially CFS-coupled signals of interest, but typically even more strongly broad-band neuronal activity. Thus, the employed SNR estimates are likely grossly overestimated. In turn, the CFS effects predicted from amplitude changes are likely underestimated. In light of this consideration, the authors' statement that predicted and observed CFS effect were not correlated is particularly important, because this would provide evidence against an SNR confound irrespective of incorrect SNR estimates. I suggest to report corresponding statistics to substantiate this claim. Furthermore, it should be clarified why the analysis is restricted to 200 parcels with strongest CFS effects.

2) The description of the new Figure 7/C seems misleading. My understanding is that the apparent match between 1:1 and CFS coupling (filling/border) is simply a necessity of the fact that only matching areas were plotted. E.g., there could not have been a region with a yellow filling and violet border in these plots. If this is indeed correct, it should be clearly stated that these figures are not used to assess the match between 1:1 and CFS networks, but merely to identify matching regions.

Reviewer #3:

The authors have responded to the comments as best as they could or were willing to (e.g., no new subjects tested). I don't see any major technical problems with the manuscript or the analyses.

My main concern is the same as with the initial submission: I'm just not sure what to make of the results. It's a bit of a "throw the spaghetti against the wall and see what sticks" approach. The brain is sufficiently complex that pretty much any all-to-all data analysis application will produce "stuff", some of which may be real and meaningful and some of which may be spurious or at least not generalizable. There are many publications in the literature that do massive all-to-all exploratory connectivity analyses, and it's never really clear what they mean, except to confirm that there are large-scale networks in the brain.

The authors' argument is that phase-amplitude coupling might not be fast enough for some of the faster aspects of cognition, and therefore the brain might be phase-phase synchronization (CFS). Fair enough, and it's hard to disagree with this claim. But I'm not sure that this experiment provides necessary evidence for this. Why does WM need to be fast? Why couldn't the brain use "slow" PAC here? (Does the brain even need PAC for WM?)

I think an important conceptual advance would come from showing that CFS "solves a problem" or at least correlates with task performance in a way that is unique amongst connectivity manifestations. Otherwise, this manuscript -- methodologically rigorous as it is -- seems to be just another example of how a massively complex system like the brain can be measured using many different analysis techniques.

I hope my comments are not interpreted as being too negative or critical. The authors clearly put a lot of work and careful thought into this manuscript. It just seems to me to be more a methodological exercise than a conceptual or theoretical advance. But that's just my opinion.

---

## [Author Response]

*Robustness of results*

1) The sample (N=12) is very small for a correlation analysis. This is compounded by the number of correlations computed. No scatterplots are shown to illustrate whether the findings may have been driven by outliers.

Overall we agree, but would like to point out that the data from individual subjects is of much higher quality than is common in the field because a total of 1800 VWM trials were acquired from each subject. We have now addressed the limitations of sample size in a few complementary ways to show that the results are robust and not driven by outliers. Furthermore, we would like to note that the manuscript has only two core CFS group analyses: one shows that CFS is stronger during the VWM retention period than in the baseline (current Figure 3) and another that CFS is modulated by VWM load (current Figure 3). The other analyses use these data in order to visualize the neuroanatomical structure of the significant CFS observations (Figure 4–Figure 5), estimate interactions between functional subsystems (Figure 6), estimate the correlation between CFS and 1:1 synchrony (new Figure 7), and estimate the correlation between CFS and VWM capacity (current Figure 9).

To consolidate the primary statistical procedure, we have now applied two controls:

1) We performed leave-one-out statistics to exclude the contribution of outliers for the main results (Mean and Load conditions, positive tail). Here, we performed the group statistics 12 times by using 11 subjects so that each subject was left out once. We then took the adjacency matrices of the original full-cohort group statistics (see Figure 3) and of the significant (FDR-corrected) observations (parcel-parcel connections) therein, we kept only those that were significant (without FDR) in every oneof the 12 leave-one-out tests. Hence, no single subject could drive the group-level observation. Despite being a drastic criterion, the effects of this pruning were minor on the overall patterns of CFS findings (Figure 3—figure supplement 2).

2) We identified the effects sizes (ES) that our cohort of *N* = 12 would have been adequate for detecting reliably and tested whether the ES observed in the data exceeded these limits. Analysis of statistical power shows that an ES of 0.9 for CFS in the Mean condition yields to statistical power 1–β = 0.8 or greater with N = 12. For the Load condition, this statistical power is given by a correlation coefficient of *r* = 0.35. To test whether our data exceeded these limits, we pruned from the original significant findings (Figure 3) all observations (connections) with ES below these limits. Hence, the remaining data contain only such CFS interactions that were sufficiently large to be considered detectable with our sample size and at 1–β = 0.8. The pruning of the low-ES findings did not lead to any major changes in the overall patterns of CFS (Figure 3—figure supplement 2) and hence the present results are unlikely to be confounded by the small sample-size. However, it is important to note that this procedure is neither to an *a* priorinor a post hocpower analysis but rather provides an empirical test for whether the observed effect sizes are in line with statistical prerequisites.

Finally, we combined both the leave-one-out and effect-size masks to obtain the most conservative estimate of significant observations in our data (Figure 3—figure supplement 2 C, F). Here we thus require a “significant” CFS connection to (i) be significant in full cohort FDR-corrected statistics, (ii) be significant in every leave-one-out sub-cohort statistic, and (iii) pass the effect size criterion. Despite this rigorous criterion, we still observed Mean- and Load-condition CFS effects very similar to those presented in Figure 3. These confirmatory analyses thus suggest that the reported findings are robust and cannot be explained by outliers. The same procedure was also performed for our PAC findings (Figure 8—figure supplement 1). These findings are reported in the Results section.

Overall, we would like to point out that even though our data-driven analysis and statistical approach is indeed based on massive amounts of statistical tests (spanning frequencies, ratios, time-windows, and parcel-parcel pairs), we control multiple comparisons rigorously. This kind of a data-driven approach is necessary for the identification of the physiological phenomena that are most robust in the data as well as for the mapping of the correlation networks at the whole-brain level. This scale analyses has also been on several occasions suggested to be crucial for understanding neuronal communication in cognitive architectures (Kriegeskorte et al., 2009; Petersen and Sporns, 2015, Deco et al., 2015; Spadone et al., 2015). Conversely, for this purpose, strictly hypothesis-driven a priori*-* (which can amount to double-dipping-) based selection of anatomical and spectral ROIs is always suboptimal as the robustness of the identified phenomenon is left unclear and the alternative explanations untested.

*It's also not very clear in Figure 8 which correlations are significant, as the asterisks seem to be on top of all four bars. There are more comparisons than subjects.*

We are sorry about the figure being incomprehensible and for the inadequate explanation in the figure caption. This figure displayed the correlation of CFS with the individual VWM capacity separately for each ratio although statistics was obtained for CFS data concatenated over ratios. The aim was to illustrate how different ratios contribute to the concatenated data.

We have now (i) modified the analysis underlying the figure to average rather than concatenate the ratios, (ii) modified the figure to show both the statistical correlation and individual ratios, and (iii) rewritten the Results, Figure caption, and the corresponding Methods section for clarity. We now show correlation values between the individual VWM capacity and the mean CFS or PAC network strength*, S*, across low- (1:2–1:5) and high- (1:6–1:9) ratios separately for each low frequency *f_low_*(current Figure 9). This shows that high-α-consistent CFS is correlated with capacity at low ratios while θ-consistent CFS is correlated with capacity at high ratios. The findings remain significant after correction for multiple comparisons with the Benjamini-Hochberg FDR procedure, which is conservative and accurate for small N. This figure thus shows *both* that the main phenomena discovered in this paper, θ- and high-α-consistent CFS, are correlated with VWM capacity *and* that such a relationship is not observed in any other low frequency or in PAC.

To illustrate without statistics the consistency of the data underlying this analysis, we also provide the correlation coefficients for each ratio in separate figure panels (Figure 9).

*2) Potential confounds due to changes in signal amplitude are not sufficiently addressed. The authors correctly point out potential confounding of cross-frequency synchrony (CFS) by changes in signal-to-noise ratio due to task-induced changes in oscillations amplitudes. However, the reported control analysis of amplitude-amplitude coupling does not address this confound, because it tests for changes in power co-variance, but not for changes in the mean power between baseline and memory intervals. However, any change in power, and thus SNR, of merely one of two analyzed nodes is expected to yield a spurious change in measured CFS even without any true change in underlying CFS. Critically, one may observe spurious decreases and increases depending on the relative strength of CFS and amplitude changes for signal and noise components. In previous analyses of the same data, the authors reported amplitude changes during visual working memory (Palva et al., J Neurosci 2011) that were load-dependent and performance-predictive just like the CFS effects reported here. It should be ruled out that the present CFS results reflect spurious effects driven by these amplitude changes. E.g., the authors should directly compare CFS and amplitude effects and may stratify the data for potential amplitude changes before quantifying CFS.*

This is a valid concern. We have earlier addressed the relationship between event-related amplitude changes, SNR, and 1:1-phase synchrony in the supplementary material for Palva et al., PNAS 2010. There we found, using the same dataset as here, that changes in 1:1 phase synchrony were poorly predicted by changes in SNR. The retention period amplitudes were often suppressed below baseline levels while synchrony was strengthened and in cases of changes to the same direction, the SNR-predicted changes of synchrony were approximately ten times smaller than the observed changes in synchrony. This disparity suggested that even the reverse cause-and-effect could be plausible: perhaps the local consequences of changes in inter-areal synchronization were driving the observed changes in amplitudes rather than vice versa.

We have now adapted this extensive control analysis to the cross-frequency context of the present study. To increase accuracy, we first obtained new SNR estimates so that instead of using sensor-space SNR (as in 2010), we now have local, parcel-by-parcel defined source-space SNR estimates. We thus have accurate SNR estimates for the baseline and retention period time windows of every parcel and frequency band separately for each VWM load condition. Hence the local SNR changes caused by the observed event-related and memory-load dependent amplitude changes could be accurately predicted. We then used the known relationship between changes in PLV and changes in SNR to predict the changes in CFS-PLV from the observed amplitude/SNR changes in these data. We found for CFS that, similarly to 1:1 phase coupling, the observed changes in PLV could not be explained by changes in SNR. Moreover, because the SNR changes were essentially uncorrelated with the PLV changes, this conclusion is robust also against systematic over- or underestimations of SNR. These findings are discussed in a new section of the Results section and illustrated in Figure 3—figure supplement 3. The rationale and methods are explained in Materials and methods section 11.

*3) The interpretation of network results seems at odds with the shown data at several points. The description of CFS networks (Figure 4; Results section) stresses the claimed link between executive (FP, DA) and visual networks. However, this does not become clear from Figure 4, Figure 5 and 6. E.g., it is stated that FP high-α connects to visual γ, or more generally, that theta and high-α consistent CFS connects the visual system with FP and DA networks. This is neither readily apparent from Figure 4 nor from Figure 6. In fact, in contrast to what is stressed in the text, there seem to be hardly any visual hubs in Figure 4.*

We acknowledge that the network illustrations were suboptimally explained and already thereby tricky to interpret. While we also see that some aspects of the results were inadequately discussed (e.g.,connectivity with SM, DM, and the CFS suppression), we feel that overall the interpretation of network results is well backed by the data. Because the CFS networks are not symmetrical, they are, frankly, quite difficult to illustrate. We have now fully rewritten the Results section, Figure captions, and the Methods section to explain more explicitly how the graphs were constructed and intended to be read. We hope that they are now more easily understandable.

Both the graph illustrations (Figure 4, Figure 5) and sub-system analyses (Figure 6) converge on showing CFS mainly among attentional and between attentional and visual systems. However, these two analyses cannot be expected to yield perfect overlap because in Figure 4 and Figure 5, the illustration is focused on most central hubs and connections and also includes a compensation for M/EEG signal mixing. Figure 6, on the other hand, uses all the data and is agnostic towards the centrality of the connections.

Another possible cause of confusion is that we present in Figure 4 and Figure 5 the network hubs and strongest connections of the low-frequency (LF) and high-frequency (HF) CFS networks in separate graphs (see, e.g., the low-frequency (LF) and high-frequency (HF) columns in Figure 4) in order to simplify both the graphs and their interpretations. With LF and HF hubs and their respective connections in separate figures, the complete picture is only obtained when looking at both of these figures and also at the line ends in addition to the hubs. We illustrate the graphs so that, in the LF-part, the hubs are the greatest LF hubs (parcels with most CFS connected low-frequency oscillations) while the dashed ends of the connection lines indicate brain area the hub is connected to, i.e., the area of the faster oscillation. The opposite scheme is then used for the HF part. For example in Figure 4, the left column (LF) shows that high-α CFS-hubs are robust in the FP (blue circles) and are strongly connected to γ oscillations in the visual system (reddish or purple edges with dash lines ending to the visual cortex). Conversely, the right column (HF) shows that γ CFS-hubs are robust in DA (yellow), DM (grey), and visual (red circles) system, and are coupled to α oscillations, e.g., in frontal cortex (dashed line ends). Hence, because these figures disambiguate both ends (low- and high frequency) of the most central CFS connections, it can be seen that CFS connects α oscillations in FP and superior frontal sulcus/gyrus with γ oscillations, e.g., in DA and visual system. Understanding the CFS network architecture would be essentially impossible if the LF and HF parts of the CFS networks were presented in the same graph

*At the same time, the sensorimotor network, which the authors refer to as putatively task-irrelevant, consistently shows up in CFS networks (Figure 6), but this is not discussed.*

We have now added discussion on this topic in subsection “The roles of CFS in sensorimotor and default mode systems”, where we write:

“fMRI studies have shown that during VWM, neuronal activity is also strengthened in the SM system wherein the dorsal premotor cortex (PMC) has been suggested to underlie the maintenance and manipulation of spatial features (Munk et al., 2002; Mohr et al., 2006). […] Together with the prior fMRI data, this supports the notion that the SM system could be task-relevant in VWM, but further studies are needed to disentangle their functional roles.”

*Similarly, the authors suggest CFS suppression of task-irrelevant SM and DM networks, but do not mention that CFS was also suppressed in putatively task-relevant DA and FP networks (Figure 6, 4th matrix). The description of networks should be clarified and inconsistencies with the hypothesis should be pointed out.*

We fully agree. The previous version did not adequately acknowledge these aspects of the results. We have now expanded discussion on this topic in subsection “Suppression of CFS in task-irrelevant networks”. We now write:

“The DM system is widely thought to support non-task-oriented processing (Fox et al., 2009), which suggests that the suppression of CFS there could represent the disengagement of task-irrelevant brain regions and networks, even though some parts of DM could be task-positive in VWM (see above). […] These results suggest that CFS can underlie both the integration as well as segregation of neuronal processing suggested to be implemented via hierarchical network interactions (Sporns, 2013; Deco et al., 2015; Spadone et al., 2015).”

*It may also help to simplify or split the network display of Figure 4 and Figure 5, which convey a lot of information, but are hard to parse.*

We agree and we have tried to mitigate this problem by first fully rewriting the Results section, Figure caption, and the Methods sections to clarify the description of these results and explanation of how the figures should be parsed. The previous submission was obviously lacking in this respect. We have also added a schematic illustration of the concept of CFS between low- and high-frequency networks as a new Figure 1, which we hope to clarify the context of the connections in Figure 4 and 5. Finally, we also have a new simplified visualization of CFS linking 1:1 networks that we hope to help here (new Figure 7, and Figure 7—figure supplement 1 and Figure 7—figure supplement 2).

With respect to the networks in Figure 4 and Figure 5; they already illustrate a single network split into two (low and high freq.) components: we would hesitate to split the graphs further and fear that a multitude of subgraphs would be even more confusing to the reader. For reasons explained above (see 3 and footnote 1), merging these graphs is not informative either. Finally, these graphs are already greatly simplified from the complete graphs and they aim to illustrate the fundamental ‘backbone’ of the real networks. We experimented with simplifying them further but this entails losing significant hubs and inter-areal connections, and thus ultimately giving way to misinterpreting the data. Nevertheless, if the Reviewers or Editors feel that Figure 4 and Figure 5 are excessive in complexity or would still benefit from further, more schematic, visualizations (perhaps akin to those in Figure 7), we are ready to provide them.

*4) The evidence supporting a link between CFS and 1:1 networks is weak (Figure 3; Results section). The reported correlations between networks seem low. Furthermore, these correlations may be driven by global gradients rather than by a match of distinct network hubs. I suggest showing CFS and 1:1 networks (degree and/or betweenness) to substantiate the suggested match.*

We have now redone these analyses to estimate separately the correlations between the low-frequency 1:1 networks hubs and low-frequency of CFS hubs, and the same for corresponding high-frequency 1:1 and CFS. The new approach revealed much stronger correlations than the previous where the low and high were mixed. The new data show that both the theta and high-α 1:1 hubs are clearly correlated with the corresponding theta and high-α CFS low-frequency hubs (new Figure 7). Similar, but not as strong, correlations were found for their high-frequency counterparts. The new analysis is more accurate and statistically more powerful than the prior, and we hope that it more convincingly provides evidence for a link between CFS and 1:1 networks.

Regarding the suggestion of showing both CFS and 1:1 networks, we have now added force-field graph representations of how the strongest hubs of the 1:1 networks are linked by CFS connections and co-localized with CFS-hubs (new Figure 7). Importantly, it shows that many of the CFS-hubs visible in the full network graphs (Figure 4) are indeed connecting 1:1 networks as hypothesized. These new results are presented in the Results section. We did not include visualizations of the full 1:1 networks with the fear of over-complicating the paper – if the reviewers feel that they would be useful, we are ready to add them as well.

We also show in separate figures (Figure 7 and Figure 7—figure supplement 2) how 1:1 high-α is connected by CFS to β, low-γ, and high-γ 1:1 networks. Importantly, this visualization reveals distinct connectivity patterns for these networks suggesting that CFS correlations were not driven by global gradients. The lack of influence of global gradients is also strongly supported by that the centrality correlations were significant regardless of the number of vertices (70-148) used in the analyses. If global gradients (locally variable MEG reconstruction accuracy) underlied these correlations, they should be maximal when all vertices are used and decay when worst vertices are excluded. However, this was not the case and using as few as the 70 best and excluding the 78 worst (best/worst in MEG connectivity reconstruction accuracy) yielded correlations comparable to those reported in Figure 7 (data not shown).

This is now discussed in the Results section in where we write:

“Despite both low- and high-γ networks being characterized by hubs in the visual system, they were not mutually significantly correlated in parcel centralities (Figure 7—figure supplement 3), which implies that these γ oscillations were relatively narrow-band and cannot, at least not entirely, be explained by broad-band γ activities”.

*5) Why were high-α consistent networks only investigated for ratios 1:3-1:9, but not for 1:2? 1:2 CFS showed mean and load effects (Figure 2). This seems like an arbitrary sub-selection.*

The graphs for 1:2 α-β CFS are shown in Figure 4—figure supplement 1. These networks were shown separately, because they were anatomically very different from those presented in Figure 4 in being largely focused into the sensorimotor system. 1:2 α-β coupling appears to have a significant contribution from the joint α and β oscillations that comprise the sensorimotor mu-rhythm. Another important reason for excluding this coupling from α-γ coupling visualizations is that functionally the β oscillations are distinct from the γ oscillations and thus α-β interactions are likely play a different functional role than α-γ CFS. This is now better explained in the Results section and also discussed in the Discussion section. We also now show high-α 1:2 CFS subsystem interaction matrices in Figure 6.

In the Results section we write:

“In contrast to α:γ CFS, 1:2 CFS connections between high-α and β oscillations were localized to the sensorimotor (SM), DA and VA networks (Figure 4—figure supplement 1, Figure 4—source data 5) and were thus visualized separately from α:γ CFS. These data showed that strongest connections of α:γ CFS are distinct from α:γCFS connections and suggested that α:γ and α:γ underlie functionally distinct CFC”.

*6) Inferential statistics (with p-values) seem to be missing for the claimed increases and decreases of individual-frequency CFS during VWM (Figure 2).*

These data were provided in the online materials accompanied with the submission ([Supplementary-material SD1-data]). We provide effect sizes and p-values for Mean and Load conditions in online materials per journal convention.

*7) No spectral graphs are shown, or comparisons of topographical maps with source reconstructions.*

We are not certain whether we correctly understand this concern. All data presented in the manuscript are based on source reconstructed MEG/EEG data – we do not show any sensor-level MEG or EEG data. Our primary inferential statistics (Figure 3) span the plane of frequencies and coupling ratios, which is the CFS counterpart for a spectral graph. Finally, the network and subgraph visualization on flattened cortical surfaces (Figure 4, Figure 5) aim to serve the role of topographical maps.

Nevertheless, we are sorry about that our prior description of the data-analysis pipeline was inadequate and unclear. We have now greatly revised the Materials and methods section and added a new figure (Figure 3—figure supplement 1) to better explain the data-analysis workflow. In this figure, we identify the raw, intermediate, and final data elements as well as the analysis processes linking them. We also provide here the links to all figures and to the corresponding sections in Materials and methods. Conversely, the workflow figure is referred to in the Materials and methods. We hope that these changes clarify the data-analysis approach and how each main result was obtained.

*8) In Figure 2, significant interactions span many n:m ratios. This seems to be a robust observation, as subsequnt figures collapse over many frequency pairs from 1:2 or 3:9. I think this is what they refer to as harmonic consistency. But the interpretation is not very clear; I think it means that the coupling is actually with fast broadband fluctuations and they are only measuring these fluctuations at a limited number of sampled frequency points. If these reflect rhythmic interactions, the authors would need to show that there are separated spectral peaks at each integer multiple. I wouldn't consider low frequencies coupled to broadband fluctuations a "confound" (I guess the authors wouldn't either), but the interpretation of phase-phase oscillatory synchronization is quite different from the interpretation of phase-broadband coupling.*

We fully agree with the reviewer’s concern of interpretational challenges. The primary motivation for showing multiple ratios collapsed into a single graph was to simply reduce the figure load and to give the reader a tangible summary of the results even though the graphs merged across ratios might not be evenly representative of each individual ratio. Overall, however, the principal network connectivity elements, and the CFS between frontal and visual cortex in particular, were similar in all collapsed networks and hence we feel that this visualization approach still gives a fair overview of the most significant phenomena in the data.

While the interpretation of fast oscillations reflecting broadband fluctuations would also be plausible, especially for the higher γ frequencies, the data appear to provide evidence against it. First, an examination of the cross-frequency similarities of the 1:1 network vertex centralities (new Figure 7—figure supplement 3) shows that while the classical frequency bands are internally fairly well clustered (similar between the nearby frequency bands), there are negligible similarities across frequency bands. In particular the low-γ (30-60 Hz) and high-γ (60-90 Hz) bands are mutually uncorrelated and if they were a single broadband phenomenon, one would expect much more widespread similarities. The low- and high-γ bands being distinct phenomena is also supported by distinct connectivity patterns of CFS with these 1:1 networks (new Figure 7 and Figure 7—figure supplement 2, Results). Hence, there appears to be no salient evidence for the γ oscillations during VWM retention being broadband activity while the band- and ratio-limited nature of these correlations speaks for genuinely oscillatory components.

*Relevance and Advance*

*This is a very rich and complicated set of results linking synchronization across different temporal scales to visual working memory performance. There are a lot of results and it is not clear how they are all related. The take-home message seems to be that the brain is complicated and produces myriad dynamics. Of course I don't doubt such a conclusion, but it is difficult to understand the specific contribution to the literature. In other words, what have we learned here?*

The systems-level neuronal mechanisms controlling the maintenance of information in VWM have remained poorly understood despite massive research efforts. It is commonly accepted that γ oscillations and inter-areal synchrony contribute to the maintenance of information in VWM. Yet, numerous studies show that also theta-, α, and β-band oscillations play a role in VWM. Rather than maintaining information *per se,* these are thought to underlie higher-level attentional and executive control of VWM. To date it has remained unclear how these oscillatory processes communicate and how their mutual information transfer may be regulated.

As postulated in the Introduction and Abstract, we hypothesize that CFS integrates the processing distributed across fast γ and slow theta / α oscillations and thereby across sensory and attentional functions of VWM. We postulate also that this integration of spectrally distributed processing is essential to achieve subjectively coherent cognition including VWM.

Unlike for PAC, there are exceedingly few papers on CFS. Our paper is the first to provide evidence for the existence of CFS in source reconstructed MEG (or EEG) data, the first to identify its networks of cortical sources, and the first to show that neocortical CFS is functionally significant VWM (or in any cognitive function). Furthermore, we show that CFS is distinct from both amplitude-amplitude and phase-amplitude cross-frequency interactions and thus an independent neurophysiological cross-frequency coupling mechanism of which the functional significance has hitherto remained unknown.

So, as to what have we learned here:

1) Our results are the first to show that CFS couples neuronal oscillations and 1:1 networks in visual and attentional systems, and that CFS therein is predictive of behavioral VWM capacity. The correlation with behavior indicates that CFS is functionally significant in VWM.

2) The cortical localization and functional significance support the idea that CFS is a novel (underacknowledged) mechanism for regulating communication and integrating neuronal processing in distinct frequency bands and functional systems. We propose that in VWM, CFS integrates processing among synchronized neuronal networks from theta to γ frequencies to link sensory and attentional functions.

We have now rewritten much of the Abstract, Introduction and the Discussion to be explicit about how our findings advance the prior literature and to better explain the putative functional role of CFS in WM.

*Readability*

*The findings are significant and important, but I'm concerned about the overall readability. I consider myself fairly fluent in these domains (CFC, graph theory) but it still took me quite some time to wrap my head around the results enough to put them into a mechanistic and cognitive context. I strongly believe that a clear schematic figure, up front, would help guide readers through the rest of the paper.*

Thank you for this suggestion. We have now added a schematic figure (new Figure 1) that presents the hypothesis and illustrates the idea of how CFS can phase-dependently connect slow and fast 1:1 networks. In a new Figure 7, we now also illustrate with simplified graphs how CFS connections link the key hubs of 1:1 networks. Finally, throughout the manuscript, we have done our best to clarify the text and figure legends. Further, we have added a schematic workflow figure (Figure 3—figure supplement 1) to better illustrate our methods.

*While I recognize that the current Figure 1 attempts to do this, there are certain design features that hinder it. For example, upon my first viewing, the fact that the colors used to differentiate cortical regions and connection in 1a (blue, purple, green, red, yellow) are the same colors chosen for the squares when illustrating a VWM trial suggests a link between the two when, in fact, they are unrelated. To be honest, my first thought was that there was so spatial decoding element to the paper.*

We apologize for this confusion. We have addressed this issue by changing the colors in the current Figure 2 (previous Figure 1) and by clarifying the figure legend.

*[Editors' note: further revisions were requested prior to acceptance, as described below.]*

*All reviewers found that the revised manuscript addresses most of their concerns.*

*I would ask you to consider Reviewer 2 comments on the SNR and description of Figure 7.*

*We hope that these issues can be dealt with expeditiously.*

We thank the editors for giving us the opportunity to revise the manuscript as well as the reviewers for their comments. We have now addressed these comments and, in particular, dealt with the SNR issue exhaustively.

*Reviewer #2:*

*The authors have extensively revised the manuscript and included several new analyses. This has substantially strengthened the manuscript and most of my concerns have been addressed. Overall, this is a very good paper reporting several intriguing and novel results. However, two points remain:*

*1) I am concerned that the SNR confound analysis is based on the invalid assumption that only the measurement noise (assessed with an empty room measurement) constitutes noise in the present context of a potential SNR confound. In contrast, for this confound, also any neuronal signals not of interest (and likely without CFS) should be considered noise. In particular, this noise entails broad band (1/f) activity, which constitutes a major part of the measured neuronal signal. For any frequency band, the measured neuronal signal does not only reflect oscillatory and potentially CFS-coupled signals of interest, but typically even more strongly broad-band neuronal activity. Thus, the employed SNR estimates are likely grossly overestimated. In turn, the CFS effects predicted from amplitude changes are likely underestimated. In light of this consideration, the authors' statement that predicted and observed CFS effect were not correlated is particularly important, because this would provide evidence against an SNR confound irrespective of incorrect SNR estimates. I suggest to report corresponding statistics to substantiate this claim. Furthermore, it should be clarified why the analysis is restricted to 200 parcels with strongest CFS effects.*

Thank you for this note. We fully agree on there being also neuronal “noise signals” in addition to the measurement noise. Using the parcel-wise amplitude measurements as a proxy for such a de factoSNR, we have now systematically mapped all correlations between task-effects on oscillation amplitudes and on the CFS PLV in the Mean and Load conditions. If SNR at large is a confounding factor, the amplitude effects should be predictive of the CFS effects in a pattern reminiscent of the one found for CFS.

We estimated the amplitude-effect/PLV-effect correlation for every parcel pair where significant CFS effects were found. Overall, even at *p <* 0.05 and without any corrections for multiple comparisons, very few significant amplitude/PLV-correlations were found. These data are now presented in the new Figure 3—figure supplement 6. In the few frequency-ratio pairs where a significant correlation was observed, the correlation explained only a small fraction of the variance in CFS PLV. Finally, the significant findings in this analysis did not form any pattern similar to the main CFS findings, which suggests that CFS is, indeed, not significantly confounded by concurrent amplitude (SNR) changes. This analysis is now explained in Methods and Results sections.

As a side note that is contextually relevant, but not for the control analyses performed here, we see that a considerable fraction of the “1/f broad band” component in brain activity recordings can be attributable to transient oscillations at variable frequencies rather than to actual 1/f-noise-like brain activity per se. Even without any power spectral peaks, CFS among brief 1/f-scaled oscillations would be a plausible phenomenon.

Finally, restricting the original SNR analysis to 200 parcel pairs with strongest CFS effects was done with the rationale that if the CFS effects were brought about by SNR changes, this relationship would be most clearly observable in the subset of data where CFS was most robust. Our aim was to use an adequate amount of high-quality data in this analysis to avoid arriving to a false negative finding. This approach is now complemented by the new PLV-amplitude correlation analyses where all significant CFS-findings are used. The corresponding methods text has been revised to clarify the rationale.

*2) The description of the new Figure 7/C seems misleading. My understanding is that the apparent match between 1:1 and CFS coupling (filling/border) is simply a necessity of the fact that only matching areas were plotted. E.g., there could not have been a region with a yellow filling and violet border in these plots. If this is indeed correct, it should be clearly stated that these figures are not used to assess the match between 1:1 and CFS networks, but merely to identify matching regions.*

We apologize for the slightly inaccurate description of this figure. The reviewer is correct in that in this visualization specifically aims to reveal which hubs of 1:1 networks are connected by CFS (and which are also CFS hubs). We have now clarified the text related to this analysis.

In subsection “*CFS connects the hubs of within-frequency (1:1) synchronized networks”* we write “We then used simplified graphs to illustrate the strongest hubs of the fast and slow 1:1 networks, their within-frequency 1:1 connectivity, and how these two networks were mutually connected by CFS. We first visualized the main hubs and their connections of 1:1 θ and α networks and the 1:2-1:3 α:γ CFS linking these networks by illustrating separately how the 1:1 θ network was CFS-connected with targets in the 1:1 α network (Figure 7, left column) and how the 1:1 α network was CFS-connected with targets in 1:1 θ network (Figure 7, right column)”.

In the same section we write now “For high-α consistent CFS, we examined which nodes of the high-α band networks were coupled through CFS with the low-β band 1:1 network…”

In the same section we also now write “For high-α consistent CFS, we examined which high-α–network hubs were coupled through CFS with the hubs of the low-γ band 1:1 network”.

*Reviewer #3:*

*The authors have responded to the comments as best as they could or were willing to (e.g., no new subjects tested). I don't see any major technical problems with the manuscript or the analyses.*

*My main concern is the same as with the initial submission: I'm just not sure what to make of the results. It's a bit of a "throw the spaghetti against the wall and see what sticks" approach. The brain is sufficiently complex that pretty much any all-to-all data analysis application will produce "stuff", some of which may be real and meaningful and some of which may be spurious or at least not generalizable. There are many publications in the literature that do massive all-to-all exploratory connectivity analyses, and it's never really clear what they mean, except to confirm that there are large-scale networks in the brain.*

*The authors' argument is that phase-amplitude coupling might not be fast enough for some of the faster aspects of cognition, and therefore the brain might be phase-phase synchronization (CFS). Fair enough, and it's hard to disagree with this claim. But I'm not sure that this experiment provides necessary evidence for this. Why does WM need to be fast? Why couldn't the brain use "slow" PAC here? (Does the brain even need PAC for WM?)*

These are all very good points. First, with respect to the caveats of the all-to-all-like neuroinformatics approach – we agree that exploratory connectivity analyses are difficult to validate for generalizability and even to illustrate and discuss in a rigorous manner. Yet, considering that brain activity in vivo crucially depends on these large-scale networks, progress must be made to understand also this level of cerebral organization. All-to-all analyses still, by definition, will provide a better account of the ‘truth’ than region-of-interest-based connectivity analyses.

However, even though we have extensively utilized the all-to-all approach here, we see this study as being very much hypothesis based: we aimed to address whether CFS is a (neurophysiological) phenomenon that plays a functional role in human working memory. Our analyses show that, considering all data, CFS is a robust phenomenon (without hypothesis-driven exclusion of any data) and, in addition, allow for the localization of the underlying networks. Nevertheless, the networks per se are not our main result – but rather the overall evidence for CFS is.

Our hypothesis is that CFS serves the integration of synchronized networks oscillating in distinct frequencies because it is the only cross-frequency coupling mechanism that allows for consistent cross-frequency spike-timing relationships. Obviously, the exact and causal roles of CFS and PAC (or of synchronization in the first place) in large scale networks and cognitive operations remain a topic requiring much further research.

*I think an important conceptual advance would come from showing that CFS "solves a problem" or at least correlates with task performance in a way that is unique amongst connectivity manifestations. Otherwise, this manuscript -- methodologically rigorous as it is -- seems to be just another example of how a massively complex system like the brain can be measured using many different analysis techniques.*

*I hope my comments are not interpreted as being too negative or critical. The authors clearly put a lot of work and careful thought into this manuscript. It just seems to me to be more a methodological exercise than a conceptual or theoretical advance. But that's just my opinion.*

We fully agree with the reviewer that much more work is needed before to fully understand the unique role of CFS in neuronal computations. We would like to note that we have shown in our manuscript for the first time that CFS can be a functionally significant mechanism for integrating neuronal processing across distinct frequencies. We think that this is a significant conceptual advance rather than a methodological exercise. To understand the roles of CFS as a general mechanism for the integration of neuronal processing across frequencies, however, will require many upcoming years of work.